# Generalized Laplacian Eigenmaps

**Hao Zhu**[†,§]    **Piotr Koniusz** [∗,§,†]
[§]Data61/CSIRO    [†]Australian National University
allenhaozhu@gmail.com, piotr.koniusz@data61.csiro.au

## Abstract

Graph contrastive learning attracts/disperses node representations for similar/dissimilar node pairs under some notion of similarity. It may be combined with a low-dimensional embedding of nodes to preserve intrinsic and structural properties of a graph. COLES, a recent graph contrastive method combines traditional graph embedding and negative sampling into one framework. COLES in fact minimizes the trace difference between the within-class scatter matrix encapsulating the graph connectivity and the total scatter matrix encapsulating negative sampling. In this paper, we propose a more essential framework for graph embedding, called Generalized Laplacian EigeNmaps (GLEN), which learns a graph representation by maximizing the rank difference between the total scatter matrix and the within-class scatter matrix, resulting in the minimum class separation guarantee. However, the rank difference minimization is an NP-hard problem. Thus, we replace the trace difference that corresponds to the difference of nuclear norms by the difference of LogDet expressions, which we argue is a more accurate surrogate for the NP-hard rank difference than the trace difference. While enjoying a lesser computational cost, the difference of LogDet terms is lower-bounded by the Affine-invariant Riemannian metric (AIRM) and upper-bounded by AIRM scaled by the factor of $\sqrt{m}$. We show on popular benchmarks/backbones that GLEN offers favourable accuracy/scalability compared to state-of-the-art baselines.

## 1    Introduction

Laplacian Eigenmaps [3] and IsoMap [36] are graph embedding methods that reduce the dimensionality of data by assuming the data exists on a low-dimensional manifold. The objective function in such models encourages node embeddings to lie near each other in the embedding space if nodes are close to each other in the original space. While the classical methods capture the related node pairs, they neglect modeling unrelated node pairs.

In contrast, modern graph embedding models such as [35, 10, 44] and Graph Contrastive Learning (GCL) [37, 56, 11, 57, 55] are unified under the (Sampled) Noise Contrastive Estimation framework, called (Sampled)NCE [27, 23]. Most of GCL methods do not incorporate the graph information into the loss but follow the setting from computer vision, *i.e.*, they assume that randomly drawn pairs should be dissimilar, whereas the original sample and its augmentations should be similar [39]. In contrast, COntrastive Laplacian EigenmapS (COLES) [55] is a framework which combines a (graph) neural network with Laplacian eigenmaps utilizing the graph Laplacian matrix within a contrastive loss. Based on the NCE framework, COLES minimizes the trace difference of Laplacians.

In this paper, we analyze the relation among within-class, between-class and total scatter matrices under the rank inequality, and prove that, under a simple assumption, the distance between any dissimilar (negative) samples would be greater/equal than the inter-class distance between their corresponding class centers. Based on such a condition, we derive GLEN, a reformulation of graph embedding into a rank difference problem, which is a more general framework than other graph

---

[∗]The corresponding author.    Code: `https://github.com/allenhaozhu/GLEN`.

36th Conference on Neural Information Processing Systems (NeurIPS 2022).

embedding frameworks, *i.e.*, under specific relaxations of the rank difference problem, we can recover different frameworks.To that end, we demonstrate how to optimize the rank difference problem with a difference of LogDet expressions, a differentiable relaxation suitable for use with (graph) neural networks. We consider other surrogates of the rank difference problem, based on the Nuclear norm, $\gamma$-nuclear norm, Schatten norm, and the Geman norm. Moreover, we provide theoretical considerations regarding the low-rank optimization and connection to the Riemannian manifold in order to interpret our approach.

In summary, our contributions are threefold:

   i. We propose a rank-based condition connecting within-class, between-class and total scatter matrices under which we provide the minimum class separation guarantee. We propose a loss function, Generalized Laplacian EigenNaps (GLEN), that realizes this condition.

  ii. As the rank difference problem is NP-hard, we consider a difference of LogDet surrogate to learn node embeddings, as opposed to the trace difference (an upper bound of the difference of LogDet terms) used by other graph embedding models. We also consider other surrogates.

 iii. We study the distance between symmetric positive (semi-)definite matrices and the LogDet-based relaxation of GLEN. While enjoying fewer computations, the difference of LogDet terms of GLEN enjoys the Affine-invariant Riemannian metric (AIRM) for a lower bound and AIRM scaled by $\sqrt{m}$ as an upper bound. We explain how GLEN connects to other graph embeddings.

## 2   Related Works

**Graph Embeddings.**   By assuming that the data lies on a low-dimensional manifold, graph embedding methods such as Laplacian Eigenmaps [3] and IsoMap [36] optimize low-dimensional data embeddings. These methods [5] construct a similarity graph by measuring the similarity of high-dimensional feature vectors and embed the nodes into a low-dimensional space.

DeepWalk [31] uses truncated random walks to explore the graph structure, and the skip-gram model for word embedding to determine the embedding vectors of nodes. By setting the walk length to one and using negative sampling [26], LINE [35] explores a similar idea with an explicit objective function while REFINE [52] imposes additional orthogonality constraints which deem REFINE extremely fast. Node2Vec [9] interpolates between breadth- and depth-first sampling. COLES [55] unifies traditional graph embedding and negative sampling by introducing a positive contrastive term that captures the graph structure, and a negative contrastive random sampling. COLES solves the trace difference problem akin to traditional graph embedding models [43]. In this paper, we propose a more general loss for graph embedding, *i.e.*, COLES solves the trace difference (Nuclear norms difference) relaxation of GLEN.

Graph embedding techniques [43] provide a general framework for dimensionality reduction such as Principal Component Analysis (PCA), Linear Discriminant Analysis (LDA), and Locality Preserving Projections (LPP) [12]. All methods within this category can be considered as solving the same problem under varying assumptions, *i.e.*, maximising the intra- and inter-class separation by optimizing the trace difference, also used in metric learning [22]. However, such a family of objective functions is not motivated by the guarantee on the minimum class separation between feature vectors from different categories. GLEN, in its purest NP-hard form, provides the minimum class separation guarantee and can be realised by several formulations depending on chosen trade-offs.

**Unsupervised Representation Learning for Graph Neural Networks (GNN).**   Unsupervised GNN training can be reconstruction-, contrastive- or diffusion-based. To train a graph encoder in an unsupervised manner, GCN [17] minimizes a reconstruction error which only considers the similarity matrix and ignores the dissimilarity information. At various scales of the graph, contrastive methods determine the positive and negative sets. For example, local-local CL and global-local CL strategies are highly popular. GraphSAGE [10], inspired by DeepWalk [31], uses the contrastive loss which encourages neighbor nodes to have similar representations, while preserving dissimilarity between representations of disparate nodes. DGI [37], inspired by Deep InfoMax (DIM) [13], uses an objective with global-local sampling strategy to maximize the Mutual Information (MI) between global and local graph embeddings. Augmented Multiscale Deep InfoMax (AMDIM) [2] maximizes MI between multiple data views. MVRLG [11] contrasts encodings from first-order neighbors and a graph diffusion. Fisher-Bures Adversary GCN [34] treats the graph as generated w.r.t. some observation noise. COSTA [50] constructs the views by injecting features with the random noise.

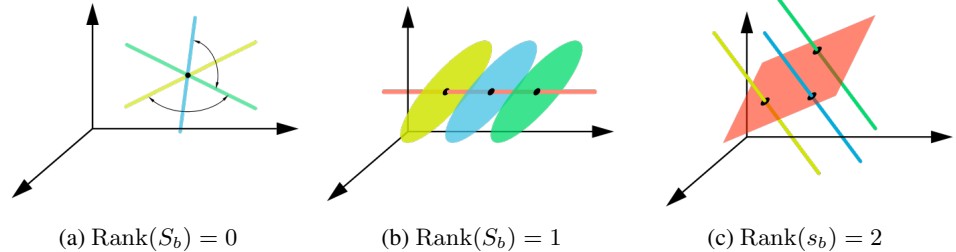

(a) $\mathrm{Rank}(S_b) = 0$  (b) $\mathrm{Rank}(S_b) = 1$  (c) $\mathrm{Rank}(s_b) = 2$

Figure 1: Three cases under our Condition 1, *i.e.*, $\mathrm{Rank}(\mathbf{S}_t) = \mathrm{Rank}(\mathbf{S}_w) + \mathrm{Rank}(\mathbf{S}_b)$. The red line and plane indicate the space of class centers. The black dots represent class centers. Other colored lines or ellipses represent the spaces of different categories. We show a non-exhaustive set of cases.

However, such contrastive approaches often require thousands of epochs to converge and perform well. In addition, many contrastive losses have an exponential increase in memory overhead w.r.t. the number of nodes. In contrast, our method does not explicitly use the local-local setting but the total scatter matrix, and thus saves computational and storage cost.

Linear GNNs, *i.e.*, SGC [42] and S$^2$GC [53], capture the neighborhood and increasingly larger neighborhoods of each node due to the diffusion, respectively. SGC and S$^2$GC have no projection layer, thus the size of embeddings is equal to the input dimension. GLEN can learn a projection layer in an unsupervised manner with a linear function or Multi-Layer Perceptron (MLP) applied to linear GNNs or any other GNN models [18, 34, 49], *etc.*

## 3 Preliminaries

**Notations.** Let $G = (V, E)$ be a simple, connected and undirected graph with $n = |V|$ nodes and $m = |E|$ edges. Let $i \in \{1, \cdots, n\}$ be the node index of $G$, and $d_j$ be the degree of node $j$ of $G$. Let $\mathbf{W}$ be the adjacency matrix, and $\mathbf{D}$ be the diagonal matrix containing degrees of nodes. Let $\mathbf{X} \in \mathbb{R}^{n \times d}$ denote the node feature matrix where each node $v$ is associated with a feature vector $\mathbf{x}_v \in \mathbb{R}^d$. Let the normalized graph Laplacian matrix be defined as $\mathbf{L} = \mathbf{I} - \tilde{\mathbf{W}} \in \mathbb{S}^n_+$, a symmetric positive semi-definite matrix and $\tilde{\mathbf{W}} = \mathbf{D}^{-1/2} \mathbf{W} \mathbf{D}^{-1/2}$. $\mathbb{S}^m_{+(+)}$ is a set of symmetric positive (semi-)definite matrices. Let $\mathbf{Z} = f_{\boldsymbol{\Theta}}(\mathbf{X}) \in \mathbb{R}^{n \times m}$ be a generalized node embedding, *i.e.*, $\mathbf{X}$ could be identity matrix (*e.g.*, no node attributes), $f_{\boldsymbol{\Theta}}(\mathbf{X})$ could be GNN or a linear function with parameters $\boldsymbol{\Theta}$. Scalars/vectors/matrices are denoted by lowercase regular/lowercase bold/uppercase bold fonts.

### 3.1 Scatter Matrices

Below are given standard definitions of scatter matrices, including the total scatter matrix $\mathbf{S}_t \in \mathbb{S}^m_{+(+)}$, the within-class matrix $\mathbf{S}_w \in \mathbb{S}^m_{+(+)}$, and between-class matrix $\mathbf{S}_b \in \mathbb{S}^m_{+(+)}$:

$$\mathbf{S}_t = \sum_{i=1}^n (\mathbf{z}_i - \bar{\mathbf{z}})(\mathbf{z}_i - \bar{\mathbf{z}})^\top = \mathbf{Z}^\top (\mathbf{I} - \tilde{\mathbf{W}}_t) \mathbf{Z} \qquad \text{where} \quad \tilde{\mathbf{W}}_t = \frac{1}{n} \mathbf{e} \mathbf{e}^\top,$$

$$\mathbf{S}_w = \sum_{i=1}^n (\mathbf{z}_i - \boldsymbol{\mu}_{y_i})(\mathbf{z}_i - \boldsymbol{\mu}_{y_i})^\top = \mathbf{Z}^\top (\mathbf{I} - \tilde{\mathbf{W}}_w) \mathbf{Z} \quad \text{where} \quad \tilde{\mathbf{W}}_w = \sum_{c=1}^C \frac{1}{n_c} \mathbf{e}^c \mathbf{e}^{c\top},$$

$$\mathbf{S}_b = \sum_{c=1}^C n_c (\boldsymbol{\mu}_c - \bar{\mathbf{z}})(\boldsymbol{\mu}_c - \bar{\mathbf{z}})^\top. \tag{1}$$

Let $\mathbf{e}$ be an $n$-dimensional vector with all coefficients equal one, $\mathbf{I}$ be an identity matrix, $\mathbf{S}_t$ be the total scatter (covariance) matrix, and $\bar{\mathbf{z}} \in \mathbb{R}^m$ be the mean of all samples. Let $\boldsymbol{\mu}_{y_i} \in \mathbb{R}^m$ be the class center of the $i$-th sample and $\boldsymbol{\mu}_c \in \mathbb{R}^m$ be the $c$-th class center. Let the total number of categories be given by $C$, whereas $n_c$ be the number of samples for the $c$-th category. Let $\mathbf{e}^c \in \mathbb{R}^n$ be a vector where a given coefficient indexed by node is equal one if its node is of class $c$, otherwise it is equal zero. We note that both $\mathbf{S}_t$ and $\mathbf{S}_w$ can take a form akin to Laplacian eigenmaps such that $\tilde{\mathbf{W}}_t$ and $\tilde{\mathbf{W}}_w$ are the corresponding normalized adjacent matrices. Let us also define graph Laplacian matrices

$\mathbf{L}_t = \mathbf{I} - \tilde{\mathbf{W}}_t \in \mathbb{S}_+^n$ and $\mathbf{L}_w = \mathbf{I} - \tilde{\mathbf{W}}_w \in \mathbb{S}_+^n$ which will be used in the sequel. Importantly, let us assume that a graph Laplacian matrix $\mathbf{L}$ containing graph links could be seen as a noisy version of $\mathbf{L}_w$ in which all nodes of a given class $c$ connect under the weight equal $1/n_c$.

Observe that $\mathbf{S}_t = \mathbf{S}_w + \mathbf{S}_b$. Thus, $\mathrm{Rank}(\mathbf{S}_t) \leq \mathrm{Rank}(\mathbf{S}_w) + \mathrm{Rank}(\mathbf{S}_b)$ due to the rank inequality. Below we highlight the condition underpinning the subsequent motivation:

**Condition 1.** $\mathrm{Rank}(\mathbf{S}_t) = \mathrm{Rank}(\mathbf{S}_w) + \mathrm{Rank}(\mathbf{S}_b)$.

### 3.2 Motivation

Figure 1 shows some three optimal solutions for Condition 1. The rank of between-class scatter matrix $\mathbf{S}_b$ for the whole dataset is at most $C - 1$ (where $C$ is the number of classes). Since $\mathrm{Rank}(AB) \leq \min(\mathrm{Rank}(A), \mathrm{Rank}(B))$, we have[†] $\mathrm{Rank}(\mathbf{S}_w^{-1}\mathbf{S}_b) \leq \mathrm{Rank}(\mathbf{S}_b) \leq C - 1$. The rank is the number of non-zero eigenvalues of a matrix so $\mathbf{S}_w^{-1}\mathbf{S}_b$ has at most $C - 1$ non-zero eigenvalues. Condition 1 implies that $\mathrm{Rank}(\mathbf{S}_w^{-1}\mathbf{S}_b) = 0$ results in the minimum class separation guarantee under that condition.

**Theorem 1.** *Let the feature dimension be larger than the class number (i.e., $m > C$) and Condition 1 hold. Then, the minimum class separation is equal to the distance between class centers. In other words, the distance between any two vectors $\mathbf{z}_i$ and $\mathbf{z}_j$ with labels $y_i \neq y_j$ is greater/equal the distance between class centers $\boldsymbol{\mu}_{y_i}$ and $\boldsymbol{\mu}_{y_j}$:*

$$\|\boldsymbol{\mu}_{y_i} - \boldsymbol{\mu}_{y_j}\|_2 \leq \|\mathbf{z}_i - \mathbf{z}_j\|_2, \ \forall y_i \neq y_j, \ i,j \in \{1, \cdots, C\}. \tag{2}$$

*Proof.* As $\mathbf{S}_w$ is the orthogonal complement of $\mathbf{S}_b$, *i.e.*, $\mathbf{S}_w^{-1}\mathbf{S}_b = \mathbf{0}$, $\mathbf{S}_w + \mathbf{S}_b = \mathbf{U}\Sigma\mathbf{U}^\top$, $\mathbf{S}_w = \mathbf{U}_{1:k}\Sigma_{1:k}\mathbf{U}_{1:k}^\top$ and $\mathbf{S}_b = \mathbf{U}_{k+1:m}\Sigma_{k+1:m}\mathbf{U}_{k+1:m}^\top$ where $1 \leq k < m$. Let $\mathbf{z}_i = \boldsymbol{\mu}_{y_i} + \mathbf{U}^\top\boldsymbol{\epsilon}_i$ where $\boldsymbol{\epsilon}_i$ is the representation under the basis $\mathbf{U}$ and $\boldsymbol{\epsilon}_{(k+1:m),i} = \mathbf{0}$ because only top k components $1:k$ represent $\mathbf{S}_w$. Thus, the orthogonal projection $\mathbf{U}_{k+1:m}$ fulfills $\|\mathbf{U}_{k+1:m}(\mathbf{z}_i - \mathbf{z}_j)\|_2 \leq \|\mathbf{z}_i - \mathbf{z}_j\|_2$. Moreover, $\mathbf{U}_{k+1:m}(\mathbf{z}_i - \boldsymbol{\mu}_{y_i}) = \mathbf{U}_{k+1:m}(\mathbf{U}^\top\boldsymbol{\epsilon}_i) = 0$. That is, all $\{\mathbf{z}_i : y_i = c\}$ are projected onto the mean $\boldsymbol{\mu}_c$. Thus, the inequality in Eq. 2 holds. □

Theorem 1 guarantees the worst inter-class distance[§]. Figure 1 shows some cases that meet Condition 1. Figure 1a shows the case for which the class centers collapse to a single point and thus the inter-class distance equals zero (collapse of the feature space). Figures 1b and 1c show other cases.

## 4 Methodology

Condition 1 points to a promising research direction in learning discriminative feature spaces. However, optimizing over the rank is NP-hard and non-differentiable. In what follows, we provide the formulation of Generalized Laplacian EigeNmaps (GLEN) and its relaxation, which is differentiable.

### 4.1 Generalized Laplacian Eigenmaps

As solving Condition 1 is NP-hard, we propose a relaxation where $\mathrm{Rank}(\mathbf{S}_t)$ is encouraged to be as large as possible (bounded by the feature dimension $m$). On the contrary, if $\mathrm{Rank}(\mathbf{S}_t) \approx \mathrm{Rank}(\mathbf{S}_b)$ then the small $\mathrm{Rank}(\mathbf{S}_w)$ limits the feature diversity. In the extreme case, if $\mathrm{Rank}(\mathbf{S}_w) = 0$, the feature representation collapses. Larger $0 < \mathrm{Rank}(\mathbf{S}_b) \leq C - 1$ improves the inter-class diversity.

> We propose a new **Generalized Laplacian EigeNmaps (GLEN)** framework for unsupervised network embedding. In the most general form, GLEN maximizes the difference of rank terms:
> $$\boldsymbol{\Theta}^* = \arg\max_{\boldsymbol{\Theta}} \ \mathrm{Rank}\big(\mathbf{S}_t\big(f_{\boldsymbol{\Theta}}(\mathbf{X})\big)\big) - \mathrm{Rank}\big(\mathbf{S}_w\big(f_{\boldsymbol{\Theta}}(\mathbf{X})\big)\big). \tag{3}$$

As the general matrix Rank Minimization Problem (RMP) [7] is NP-hard and so is the difference of rank terms in Eq. 3, we relax this problem by the difference of LogDet terms that serve as a surrogate of the NP-hard problem. Appendix I derives GLEN from the SampledNCE framework.

---

[†]We write $\mathbf{S}_w^{-1}$ but if $\mathbf{S}_w$ is rank-deficient, $^{-1}$ is replaced with the Moore–Penrose inverse (pseudo-inverse).

[§]Other graph embedding models that maximize/minimize inter-/intra-class distances have no such guarantees.

> **GLEN (LogDet relaxation).**
>
> I. Define:
> $$\delta(\mathbf{S}_t, \mathbf{S}_w; \alpha, \lambda) = \log \det(\mathbf{I} + \alpha \mathbf{S}_t) - \lambda \log \det(\mathbf{I} + \alpha \mathbf{S}_w), \tag{4}$$
> where $\lambda \geq 0$ controls the impact of $\log \det(\mathbf{S}_w)$. If $\lambda = 0$, $\delta(\cdot)$ encourages $\mathrm{Rank}(f_\Theta(\mathbf{X})) = m$.
>
> II. Let $\mathbf{S}_t = f_\Theta(\mathbf{X})^\top \mathbf{L}_t f_\Theta(\mathbf{X})$ and $\mathbf{S}_w = f_\Theta(\mathbf{X})^\top \mathbf{L}_w f_\Theta(\mathbf{X})$. Then the LogDet relaxation becomes:
> $$\Theta^* = \underset{\Theta}{\arg\max} \ \log \det \left( \mathbf{I} + \alpha f_\Theta(\mathbf{X})^\top \mathbf{L}_t f_\Theta(\mathbf{X}) \right) - \log \det \left( \mathbf{I} + \alpha f_\Theta(\mathbf{X})^\top \mathbf{L}_w f_\Theta(\mathbf{X}) \right), \tag{5}$$
> where $\mathbf{I}$ ensures $\mathbf{I} + \alpha f_\Theta(\mathbf{X})^\top \mathbf{L} f_\Theta(\mathbf{X}) > 0$ as $f_\Theta(\mathbf{X})^\top \mathbf{L} f_\Theta(\mathbf{X})$ may be $\mathbb{S}_+^m$ leading to $\det(f_\Theta(\mathbf{X})^\top \mathbf{L} f_\Theta(\mathbf{X})) = 0$. Thus, we use $\log \det(\mathbf{I} + \alpha \mathbf{S})$ as a smooth surrogate for $\mathrm{Rank}(\mathbf{S})$.

**Proposition 1.** *Let $\sigma(\mathbf{S})$ be the vector of eigenvalues of matrix $\mathbf{S} \in \mathbb{S}_{+(+)}^m$, and $\mathrm{Eig}(\mathbf{S})$ be a diagonal matrix with $\sigma(\mathbf{S})$ as its diagonal. Let $\mathbf{S}, \mathbf{S}' \in \mathbb{S}_{+(+)}^m$ and $\alpha > 0$. Then, $\delta(\mathbf{S}, \mathbf{S}'; \alpha, \lambda) = \delta(\mathrm{Eig}(\mathbf{S}), \mathrm{Eig}(\mathbf{S}'); \alpha, \lambda)$, i.e., $\delta(\cdot)$ depends on eigenvalues rather than eigenvectors of $\mathbf{S}$ and $\mathbf{S}'$.*

*Proof.* The proof follows from the equality $\det(\mathbf{I} + \alpha \mathbf{S}) = \prod_i \sigma_i(\mathbf{I} + \alpha \mathbf{S}) = \prod_i (1 + \alpha \sigma_i(\mathbf{S})) = \det(\mathbf{I} + \alpha \, \mathrm{Eig}(\mathbf{S}))$. Thus $\delta(\mathbf{S}, \mathbf{S}'; \alpha, \lambda) = \log \det(\mathbf{I} + \alpha \mathbf{S}) - \lambda \log \det(\mathbf{I} + \alpha \mathbf{S}') = \log \det(\mathbf{I} + \alpha \, \mathrm{Eig}(\mathbf{S})) - \lambda \log \det(\mathbf{I} + \alpha \, \mathrm{Eig}(\mathbf{S}')) = \delta(\mathrm{Eig}(\mathbf{S}), \mathrm{Eig}(\mathbf{S}'); \alpha, \lambda)$. □

## 5 Theoretical Analysis

Below, we compare our approach and other methods by looking at (i) the low-rank optimization and (ii) the non-Euclidean distances between symmetric positive (semi-)definite matrices.

### 5.1 Nuclear Norm *vs*. LogDet for Rank Minimization

**Claim 1.** *COLES [55] is a convex relaxation (using the nuclear norm) of the rank difference in Eq. 3:*

$$\Theta^* = \underset{\Theta}{\arg\max} \, \mathrm{Tr} \left( f_\Theta(\mathbf{X})^\top \mathbf{L}_t f_\Theta(\mathbf{X}) \right) - \lambda \, \mathrm{Tr} \left( f_\Theta(\mathbf{X})^\top \mathbf{L}_w f_\Theta(\mathbf{X}) \right) \quad \text{s.t.} \ \ \Omega(f_\Theta(\mathbf{X})) = B, \tag{6}$$

*where* $\mathrm{Tr} \left( f_\Theta(\mathbf{X})^\top \mathbf{L}_t f_\Theta(\mathbf{X}) \right) = \|\mathbf{S}_t\|_*$ *and* $\mathrm{Tr} \left( f_\Theta(\mathbf{X})^\top \mathbf{L}_w f_\Theta(\mathbf{X}) \right) = \|\mathbf{S}_w\|_*$.

The nuclear norm $\| \cdot \|_*$ can be regarded as the $\ell_1$ norm over singular values. As the $\ell_1$ norm induces sparsity, the nuclear norm encourages sparse singular values leading to low-rank solutions. If $f_\Theta(\mathbf{X})^\top f_\Theta(\mathbf{X})$ is restricted to be diagonal, $\|f_\Theta(\mathbf{X})^\top f_\Theta(\mathbf{X})\|_* = \| \mathrm{Diag} \left( f_\Theta(\mathbf{X})^\top f_\Theta(\mathbf{X}) \right)\|_1$ and the nuclear norm surrogate for the rank minimization reduces to the $\ell_1$ norm surrogate for the cardinality (rank) minimization. However, for the $m$-dimensional embedding, the solution of trace difference lies on a subspace of dimension less than $m - 1$ [3]. Thus, the constraint $\Omega(f_\Theta(\mathbf{X})) = B$ prevents the dimensional collapse, *i.e.*, $f_\Theta(\mathbf{X})^\top f_\Theta(\mathbf{X}) = \mathbf{I}$.

Compared with the trace-based relaxation, LogDet is more suitable for cardinality minimization as it is less sensitive to large singular values. Also, the difference of LogDet terms does not require decorrelation of features to prevent the dimensional collapse. We discuss this matter in Appendix A. In our case, the difference of LogDet terms is always bounded by the difference of trace terms as follows.

**Proposition 2.** *Given an embedding matrix $f_\Theta(\mathbf{X}) \in \mathbb{R}^{n \times m}$, a fixed small constant $\alpha > 0$, we have the following inequality:*
$$\log \det (\mathbf{I} + \alpha \mathbf{S}_t) - \log \det (\mathbf{I} + \alpha \mathbf{S}_w) < \alpha \, \mathrm{Tr}(\mathbf{S}_t - \mathbf{S}_w). \tag{7}$$

*Proof.*
$$\log \det (\mathbf{I} + \alpha \mathbf{S}_t) - \log \det (\mathbf{I} + \alpha \mathbf{S}_w) = \log \det (\mathbf{I} + \alpha \, \mathrm{Eig}(\mathbf{S}_t)) - \log \det (\mathbf{I} + \alpha \, \mathrm{Eig}(\mathbf{S}_w))$$
$$= \mathrm{Tr} \left( \log(\mathbf{I} + \alpha \, \mathrm{Eig}(\mathbf{S}_t) - \log(\mathbf{I} + \alpha \, \mathrm{Eig}(\mathbf{S}_w)) < \alpha \, \mathrm{Tr}(\mathbf{S}_t - \mathbf{S}_w). \tag{8}$$
□

Proposition 2 is also related to the inequality $\mathrm{Rank}(\mathbf{S}) \leq \log \det(\mathbf{I} + \mathbf{S}) \leq \mathrm{Tr}(\mathbf{S})$ [7].

## 5.2 Distance between Symmetric Positive (Semi-)Definite Matrices.

Below, we provide a perspective on non-Euclidean distances between matrices from $\mathbb{S}^m_{+(+)}$ to compare the proposed method with other graph embeddings, *e.g.*, Laplacian Eigenmaps [3] and COLES [55]. For clarity, we also reformulate the Laplacian eigenmaps and COLES into forms in Prop. 3 and 4.

**Proposition 3.** *Laplacian Eigenmaps [3] method equals to maximizing the Frobenius norm:*

$$\Theta^* = \arg\max_{\Theta} \|f_\Theta(\mathbf{X})f_\Theta(\mathbf{X})^\top - \mathbf{L}_w\|_F^2, \quad s.t. \quad f_\Theta(\mathbf{X})^\top f_\Theta(\mathbf{X}) = \mathbf{I}. \tag{9}$$

**Proposition 4.** *Contrastive Laplacian Eigenmaps [55] equals to maximizing the difference of Frobenius norm terms:*

$$\Theta^* = \arg\max_{\Theta} \|f_\Theta(\mathbf{X})f_\Theta(\mathbf{X})^\top - \mathbf{L}_w\|_F^2 - \|f_\Theta(\mathbf{X})f_\Theta(\mathbf{X})^\top - \mathbf{L}_t\|_F^2, \; s.t. \; f_\Theta(\mathbf{X})^\top f_\Theta(\mathbf{X}) = \mathbf{I}. \tag{10}$$

*Proof.*

$$\|f_\Theta(\mathbf{X})f_\Theta(\mathbf{X})^\top - \mathbf{L}\|_F^2 = \text{Tr}(f_\Theta(\mathbf{X})f_\Theta(\mathbf{X})^\top f_\Theta(\mathbf{X})f_\Theta(\mathbf{X})^\top - 2f_\Theta(\mathbf{X})\mathbf{L}f_\Theta(\mathbf{X})^\top + \mathbf{L}^\top\mathbf{L})$$
$$= constant - 2\text{Tr}(f_\Theta(\mathbf{X})\mathbf{L}f_\Theta(\mathbf{X})^\top) \geq 0. \tag{11}$$

$\square$

Note that Eq. 9 encourages the linear kernel matrix $f_\Theta(\mathbf{X})f_\Theta(\mathbf{X})^\top$ to be close to $\tilde{\mathbf{W}}_w$ while Eq. 10 encourage the linear kernel matrix to be far from the $\tilde{\mathbf{W}}_w$ at the same time.

Our loss follows the non-Euclidean geometry. Below, we demonstrate the relation of Eq. 4 to the Affine-invariant Riemannian metric (AIRM). Indeed, our loss function is bounded from both sides by AIRM and AIRM scaled by $\sqrt{m}$ respectively.

**Proposition 5.** *Let $\sigma(\mathbf{S})$ be the vector of eigenvalues of $\mathbf{S}$, for any matrix $\mathbf{S}_t, \mathbf{S}_w \in \mathbb{S}^m_{+(+)}$, we have:*

$$\|\log((\mathbf{I} + \mathbf{S}_t)^{-1/2}(\mathbf{I} + \mathbf{S}_w)(\mathbf{I} + \mathbf{S}_t)^{-1/2})\|_F \leq \log\det(\mathbf{I} + \mathbf{S}_t) - \log\det(\mathbf{I} + \mathbf{S}_w)$$
$$\leq \sqrt{m}\|\log((\mathbf{I} + \mathbf{S}_t)^{-1/2}(\mathbf{I} + \mathbf{S}_w)(\mathbf{I} + \mathbf{S}_t)^{-1/2})\|_F. \tag{12}$$

*Proof.* Given $\mathbf{A} = \mathbf{I} + \mathbf{S}_t$ and $\mathbf{B} = \mathbf{I} + \mathbf{S}_w$, we have:

$$\log\det(\mathbf{A}) - \log\det(\mathbf{B}) = \log(\det(\mathbf{A})\det(\mathbf{B}^{-1})) = \log(\det(\mathbf{A})\det(\mathbf{B}^{-1/2})\det(\mathbf{B}^{-1/2}))$$
$$= \text{Tr}\log(\mathbf{B}^{-1/2}\mathbf{A}\mathbf{B}^{-1/2}). \tag{13}$$

We have $\text{Tr}(\mathbf{A}) = \|\sigma(\mathbf{A})\|_1$, $\|\mathbf{A}\|_F = \|\sigma(\mathbf{A})\|_2$ and $\|\mathbf{x}\|_2 \leq \|\mathbf{x}\|_1 \leq \sqrt{m}\|\mathbf{x}\|_2$. $\square$

Thus, Eq. 4 is trying to find a mapping function maximizing an approximation of AIRM distance between the total scatter matrix and the within-class matrix.

## 5.3 Relationship of the LogDet model to the Schatten norm

Below we demonstrate the relationship between the LogDet, Trace and Rank operators, respectively, under the Schatten norm [28] framework. Essential is the following family of objective functions:

$$f_{\alpha,\gamma}(\mathbf{S}) = \frac{1}{c}\sum_{i=1}^m \log\left(\alpha\sigma_i(\mathbf{S}) + \gamma\right) = \log\det\left(\alpha\mathbf{S} + \gamma I\right), \quad \alpha, \gamma \geq 0, \tag{14}$$

where $\sigma_i(\mathbf{S}), i = 1, \ldots, m$, are the eigenvalues of either $\mathbf{S}_t \in \mathbb{S}^m_{+(+)}$ or $\mathbf{S}_w \in \mathbb{S}^m_{+(+)}$, which are the total scatter matrix and the within scatter matrix from our experiments, respectively. Moreover, we define a normalization constant $c$ where $c = 1$ or $c = \log(\alpha + \gamma)$ as detailed below.

Given $c = 1$, we have:

$$\lim_{p \to 0} \frac{S^p_{\gamma,p}(\mathbf{S}) - m}{p} = f_{1,\gamma}(\mathbf{S}) \quad \text{where} \quad S_{\gamma,p}(\mathbf{S}) = \left(\sum_{i=1}^m (\sigma_i(\mathbf{S}) + \gamma)^p\right)^{1/p}. \tag{15}$$

From the asymptotic analysis, we conclude that the LogDet is an arbitrarily accurate rational approximation of $\ell_0$ (the so-called pseudo-norm counting non-zero elements) over the eigenvalues of $\mathbf{S}$.

The case $p = 1$ yields the nuclear norm (trace) which makes the 'smoothed' rank difference of GLEN become equivalent of COLES. The opposing limit case, denoted as $p = 0$ recovers the LogDet formula.

One can also recover the exact Rank from the LogDet formulation by:

$$\lim_{\alpha \to \infty} f_{\alpha,1}(\mathbf{S}) = \text{Rank}(\mathbf{S}) \quad \text{if} \quad c = \log(1 + \alpha). \tag{16}$$

This is apparent because:

$$\lim_{\alpha \to \infty} \frac{\log(1 + \alpha\sigma_i)}{\log(1 + \alpha)} = 1 \quad \text{if} \quad \sigma_i > 0 \quad \text{and} \quad \lim_{\alpha \to \infty} \frac{\log(1 + \alpha\sigma_i)}{\log(1 + \alpha)} = 0 \quad \text{if} \quad \sigma_i = 0. \tag{17}$$

## 6 Experiments

We evaluate GLEN (its relaxation) on transductive and inductive node classification tasks and node clustering. GLEN is compared to popular unsupervised, contrastive, and (semi-)supervised approaches. Except for the classifier, unsupervised models do not use labels. To learn similarity/dissimilarity, contrastive models employ the contrastive setting. Labels are used to train the projection layer and classifier in semi-supervised models. A fraction of nodes (*i.e.*, 5 or 20 per class) used for training are labeled for semi-supervised setting. A SoftMax classifier is used for (semi-)supervised models, while a logistic regression classifier is used for unsupervised and contrastive approaches. See Appendix E for implementation details.

**Datasets.** GLEN is evaluated on four citation networks: Cora, Citeseer, Pubmed, Cora Full [17, 4] for transductive setting. We also employ the large scale Ogbn-arxiv from OGB [14]. See Appendix D for details of datasets.

**Metrics.** As fixed data splits [45] often on transductive models benefit models that overfit, we average results over 50 random splits for each dataset. We evaluate performance for 5 and 20 samples per class. Nonetheless, we also evaluate our model on the standard splits.

**Baseline models.** We group baseline models into unsupervised, contrastive and (semi-)supervised methods, and implement them in the same framework/testbed. Contrastive methods include Deep-Walk [31], GCN+SampledNCE developed as an alternative to GraphSAGE+SampledNCE [10],

Table 1: Mean classification accuracy (%) and the standard dev. over 50 random splits. Numbers of labeled samples per class are in parentheses. The best accuracy per column is in bold. Models are organized into semi-supervised, contrastive and unsupervised groups. OOM means out of memory.

| | Method | Cora (5) | Cora (20) | Citeseer (5) | Citeseer (20) | Pubmed (5) | Pubmed (20) | Cora Full (5) | Cora Full (20) |
|---|---|---|---|---|---|---|---|---|---|
| Semi-supervised | GCN | 67.5±4.8 | 79.4±1.6 | 57.7±4.7 | 69.4±1.4 | 65.4±5.2 | 77.2±2.1 | 49.3±1.8 | 61.5±0.5 |
| | GAT | 71.2±3.5 | 79.6±1.5 | 54.9±5.0 | 69.1±1.5 | 65.5±4.6 | 75.4±2.3 | 43.9±1.5 | 56.9±0.6 |
| | MixHop | 67.9±5.7 | 80.0±1.4 | 54.5±4.3 | 67.1±2.0 | 64.4±5.6 | 75.7±2.7 | 47.5±1.5 | 61.0±0.7 |
| Contrastive | DeepWalk | 60.3±4.0 | 70.5±1.9 | 38.3±2.9 | 45.6±2.0 | 60.3±5.6 | 70.8±2.6 | 38.9±1.4 | 51.1±0.7 |
| | GCN+SampledNCE | 61.3±4.3 | 74.3±1.6 | 42.3±3.4 | 56.8±1.9 | 60.9±5.7 | 70.3±2.5 | 32.7±1.9 | 45.2±0.9 |
| | SAGE+SampledNCE | 65.0±3.5 | 73.8±1.5 | 48.0±3.5 | 56.5±1.6 | 64.1±6.1 | 74.6±1.9 | 35.0±1.4 | 43.6±0.6 |
| | Graph2Gauss | 72.7±2.0 | 76.2±1.1 | 60.7±3.5 | 65.7±1.5 | 67.6±3.9 | 74.1±2.1 | 38.9±1.3 | 49.3±0.5 |
| | SCE | 74.3±2.7 | 80.2±1.1 | 65.4±2.9 | 70.7±1.2 | 65.7±6.0 | 75.8±2.2 | 50.7±1.5 | 60.6±0.6 |
| | DGI | 72.9±4.0 | 78.1±1.8 | 65.7±3.6 | 71.1±1.1 | 65.3±5.7 | 73.9±2.3 | 50.5±1.4 | 58.4±0.6 |
| | COLES-GCN | 73.8±3.4 | 80.8±1.3 | 66.0±2.6 | 69.0±1.3 | 62.7±4.6 | 72.7±2.1 | 47.3±1.5 | 58.9±0.5 |
| | COLES-GCN (Stiefel) | 75.0±3.4 | 81.0±1.3 | 67.9±2.3 | 71.7±0.9 | 62.6±5.0 | 73.2±2.6 | 47.6±1.2 | 59.2±0.5 |
| | COLES-S$^2$GC | 76.5±2.6 | 81.5±1.2 | 67.5±2.2 | 71.3±1.0 | 66.0±5.2 | 77.4±1.9 | 51.2±1.4 | 61.8±0.5 |
| | GLEN-GCN | 77.5±2.6 | 82.7±1.2 | 67.6±2.6 | 72.0±0.9 | 68.7±5.7 | 78.2±2.4 | 52.7±1.5 | 62.0±0.5 |
| | GLEN-S$^2$GC | **78.2±2.4** | **83.0±1.0** | **69.1±2.1** | **72.3±0.9** | **70.6±3.9** | **80.1±1.9** | **53.0±1.5** | **62.6±0.5** |
| Contrastive + Multiview | GraphCL | 72.6±4.2 | 78.3±1.7 | 65.6±3.0 | 71.1±0.8 | OOM | OOM | OOM | OOM |
| | GRACE | 64.9±4.2 | 73.9±1.6 | 61.8±3.9 | 68.4±1.6 | OOM | OOM | OOM | OOM |
| | GCA | 61.5±4.9 | 75.8±1.9 | 43.2±3.6 | 55.7±1.9 | OOM | OOM | OOM | OOM |
| Unsupervised | SGC | 63.9±5.4 | 78.3±1.9 | 59.5±3.4 | 69.8±1.4 | 65.8±4.4 | 76.3±2.3 | 46.0±2.2 | 57.7±1.2 |
| | S$^2$GC | 71.4±4.4 | 81.3±1.2 | 60.3±4.0 | 69.5±1.2 | 67.6±4.2 | 73.3±2.0 | 41.8±1.7 | 60.0±0.5 |
| | PCA-S$^2$GC | 72.1±3.8 | 81.2±1.3 | 61.0±3.5 | 68.8±1.3 | 67.5±4.3 | 73.2±2.0 | 42.3±1.7 | 59.3±0.6 |
| | RP-S$^2$GC | 65.9±4.6 | 78.1±1.2 | 51.4±3.2 | 61.7±1.6 | 66.1±5.0 | 72.5±1.9 | 31.5±1.4 | 48.7±0.6 |

Graph2Gauss [4], SCE [47], DGI [37], GRACE [56], GCA [57], GraphCL [46] and COLES [55], which are our main competitors. Note that GRACE, GCA and GraphCL are based on multi-view and data augmentation, and GraphCL is mainly intended for graph classification. We do not study graph classification as it requires advanced node pooling with mixed- or high-order statistics [40, 19, 20]. We compare results with representative (semi-)supervised GCN [17], GAT [37] and MixHop [1] models. SGC and $S^2$GC are unsupervised spectral filter networks. They do not have any learnable parameters. COLES and GLEN could be regarded as dimension reduction techniques for SGC and $S^2$GC, thus we compare them to PCA-$S^2$GC and RP-$S^2$GC, which use PCA and random projections to obtain the projection layer. We set hyperparameters based on the settings described in prior papers.

## 6.1 Transductive Learning

In this section, we consider transductive learning where all nodes are available in the training process.

**COLES *vs*. GLEN.** Table 1 shows the performance of GLEN *vs*. COLES on two different backbones, *i.e*., GCN and $S^2$GC. On both backbones, GLEN shows non-trivial improvements on all four datasets. GLEN-$S^2$GC outperforms the COLES by up to 4.6%. Table 2 evaluates GLEN on Cora, Citeseer, PubMed on the standard splits instead of the random splits. See Appendix G for comparisons to additional contrastive learning frameworks.

**Contrastive Embedding Baselines *vs*. GLEN.** Table 1 shows that GLEN-GCN and GLEN-$S^2$GC outperform unsupervised models. In particular, GLEN-GCN outperforms GCN+SampledNCE on all four datasets, which shows that GLEN has an advantage over the SampledNCE framework. In addition, GLEN-$S^2$GC outperforms the best contrastive baseline DGI by up to 3.4%. On Cora with 5 training samples, GLEN-$S^2$GC outperforms $S^2$GC by 6.8%. Finally, Table 3 shows that GLEN-$S^2$GC (small number of trainable parameters) outperforms other methods on the challenging Ogbn-arxiv.

**Semi-supervised GNNs *vs*. GLEN.** Table 1 shows that the contrastive GCN baselines perform worse than semi-supervised variants, especially when 20 labeled samples per class are available. In contrast, GLEN-GCN outperformed the semi-supervised GCN on Cora by 10% and 3.4% given 5 and 20 labeled samples per class. GLEN-GCN also outperforms GCN on Citeseer and Pubmed by 9.9% and 5.2% given 5 labeled samples per class. These results show the superiority of GLEN on four datasets when the number of samples per class is 5. Even for 20 labeled samples per class, GLEN-$S^2$GC outperforms the best semi-supervised baselines on all four datasets *e.g*., by 3.3% on Cora. Semi-supervised models (*e.g*., GAT and MixHop) are affected by the low number of labeled samples, which is consistent with [25]. The accuracy of GLEN-GCN and GLEN-$S^2$GC is unaffected.

**Unsupervised GNNs *vs*. GLEN.** SGC and $S^2$GC are unsupervised linear networks based on spectral filters which do not use labels (except for the classifier). As a dimension reduction method, GLEN helps both methods reduce the dimension and achieve discriminative features. Table 1 shows that GLEN-$S^2$GC outperforms RP-$S^2$GC and PCA-$S^2$GC under the same projection size. GLEN-$S^2$GC also outperforms the unsupervised $S^2$GC baseline (high-dimensional representation).

Table 2: Comparison with other methods on Cora, Citeseer and PubMed on standard splits.

| | Cora | Citeseer | Pubmed |
|---|---|---|---|
| GCN | 81.5 | 70.3 | 79.0 |
| GAT | 83.0 | 72.5 | 79.0 |
| DeepWalk+F | 77.36 | 64.30 | 69.65 |
| Node2vec+F | 75.44 | 63.22 | 70.60 |
| GAE | 73.68 | 58.21 | 76.16 |
| VGAE | 77.44 | 59.53 | 78.00 |
| DGI | 81.26 | 69.50 | 77.70 |
| GRACE | 81.9 | 71.2 | 80.6 |
| GraphCL | 81.89 | 68.40 | OOM |
| GMI | 80.28 | 65.99 | OOM |
| COLES-GNN | 81.9 | 70.3 | 79.1 |
| GLEN-$S^2$GC | **85.10** | **71.90** | **80.72** |

Table 3: Mean classification accuracy (%) and the standard dev. over 10 runs on Ogbn-arxiv. Results of other models are from original papers.

| Method | | Test Acc. | #Params |
|---|---|---|---|
| MLP | | 55.50±0.23 | 110,120 |
| Node2Vec | [9] | 70.07±0.13 | 21,818,792 |
| GraphZoom | [6] | 71.18±0.18 | 8,963,624 |
| C&S | [15] | 71.26±0.01 | 5,160 |
| SAGE-mean | [10] | 71.49±0.27 | 218,664 |
| GCN | [17] | 71.74±0.29 | 142,888 |
| DeeperGCN | [24] | 71.92±0.17 | 491,176 |
| SIGN | [33] | 71.95±0.11 | 3,566,128 |
| FrameLet | [51] | 71.97±0.12 | 1,633,183 |
| $S^2$GC | [53] | 72.01±0.25 | 110,120 |
| COLES-$S^2$GC | [55] | 72.48±0.25 | 110,120 |
| GLEN-$S^2$GC | | **72.67±0.26** | 110,120 |

Table 4: The clustering performance on Cora, Citeseer and Pubmed.

| Method | Input | Cora | | | Citeseer | | | Pubmed | | |
|---|---|---|---|---|---|---|---|---|---|---|
| | | Acc% | NMI% | F1% | Acc% | NMI% | F1% | Acc% | NMI% | F1% |
| k-means | Feature | 34.65 | 16.73 | 25.42 | 38.49 | 17.02 | 30.47 | 57.32 | 29.12 | 57.35 |
| Spectral-f | Feature | 36.26 | 15.09 | 25.64 | 46.23 | 21.19 | 33.70 | 59.91 | 32.55 | 58.61 |
| Spectral-g | Graph | 34.19 | 19.49 | 30.17 | 25.91 | 11.84 | 29.48 | 39.74 | 3.46 | 51.97 |
| DeepWalk | Graph | 46.74 | 31.75 | 38.06 | 36.15 | 9.66 | 26.70 | 61.86 | 16.71 | 47.06 |
| GAE | Both | 53.25 | 40.69 | 41.97 | 41.26 | 18.34 | 29.13 | 64.08 | 22.97 | 49.26 |
| VGAE | Both | 55.95 | 38.45 | 41.50 | 44.38 | 22.71 | 31.88 | 65.48 | 25.09 | 50.95 |
| ARGE | Both | 64.00 | 44.90 | 61.90 | 57.30 | 35.00 | 54.60 | 59.12 | 23.17 | 58.41 |
| ARVGE | Both | 62.66 | 45.28 | 62.15 | 54.40 | 26.10 | 52.90 | 58.22 | 20.62 | 23.04 |
| GCN | Both | 59.05 | 43.06 | 59.38 | 45.97 | 20.08 | 45.57 | 61.88 | 25.48 | 60.70 |
| SGC | Both | 62.87 | 50.05 | 58.60 | 52.77 | 32.90 | 63.90 | 69.09 | 31.64 | 68.45 |
| S$^2$GC | Both | 68.96 | 54.22 | 65.43 | 69.11 | 42.87 | 64.65 | 68.18 | 31.82 | 67.81 |
| COLES-GCN | Both | 60.74 | 45.49 | 59.33 | 63.28 | 37.54 | 59.17 | 63.46 | 25.73 | 63.42 |
| COLES-GCN (Stiefel) | Both | 62.46 | 47.01 | 59.38 | 65.17 | 38.90 | 60.85 | 63.56 | 25.81 | 63.58 |
| COLES-S$^2$GC | Both | 69.70 | 55.35 | 63.06 | 69.20 | 44.41 | 64.70 | 68.76 | 33.42 | 68.12 |
| GLEN-GCN | Both | 69.27 | 55.34 | 62.14 | 68.25 | 42.98 | 64.10 | 64.64 | 30.08 | 64.12 |
| GLEN-S$^2$GC | Both | **71.01** | **56.69** | **69.01** | **69.89** | **45.37** | **65.70** | **69.62** | **34.97** | **69.33** |

Table 5: Mean classification accuracy (%) and the standard dev. over 50 random splits. Numbers of labeled samples per class are in parentheses. The best accuracy per column is in bold. Models are organized into semi-supervised, contrastive and unsupervised groups. OOM means out of memory.

| Method | Cora | | Citeseer | | Pubmed | | Cora Full | |
|---|---|---|---|---|---|---|---|---|
| | (5) | (20) | (5) | (20) | (5) | (20) | (5) | (20) |
| GLEN (Nuclear Norm) | 76.5±2.6 | 81.5±1.2 | 67.5±2.2 | 71.3±1.0 | 66.0±5.2 | 77.4±1.9 | 50.8±1.4 | 61.8±0.5 |
| GLEN ($\gamma$-nuclear norm) | 71.8±3.0 | 77.6±1.3 | 63.2±3.1 | 69.3±0.8 | 71.2±4.3 | 78.1±1.5 | 49.2±1.4 | 60.6±0.6 |
| GLEN ($S_p$ norm) | 75.2±3.5 | 80.7±1.2 | 64.7±2.4 | 70.9±0.9 | 65.9±5.5 | 73.9±2.4 | 48.0±1.6 | 59.7±1.6 |
| GLEN (Geman norm) | 72.3±2.5 | 77.2±1.3 | 65.4±2.2 | 70.7±0.8 | 72.6±4.5 | 78.3±1.5 | 49.2±1.5 | 60.6±1.6 |
| GLEN (LogDet) | 78.2±2.4 | 83.0±1.0 | 69.1±2.1 | 72.3±0.9 | 70.6±3.9 | 80.1±1.9 | 53.0±1.5 | 62.6±0.5 |

## 6.2 Node Clustering

We compare GLEN-GCN and GLEN-S$^2$GC with three types of clustering methods:

i. Methods that use only node features *e.g.*, k-means and spectral clustering (spectral-f) construct a similarity matrix with the node features by a linear kernel.

ii. Structural clustering methods that only use the graph structure: spectral clustering (spectral-g) that takes the graph adjacency matrix as the similarity matrix, and DeepWalk [31].

iii. Attributed graph clustering methods that use node features and the graph: Graph Autoencoder (GAE), Graph Variational Autoencoder (VGAE) [17], Adversarially Regularized Graph Autoencoder (ARGE), Var. Graph Autoencoder (ARVGE) [30], SGC [42] , S$^2$GC [53], COLES [55].

We measure and report the clustering Accuracy (Acc), Normalized Mutual Information (NMI) and macro F1-score (F1). We run each method 10 times on Cora, CiteSeer and PubMed. We set the number of propagation steps to 8 for SGC, S$^2$GC, COLES-S$^2$GC and COLES-S$^2$GC following [48]. Table 4 shows that GLEN-S$^2$GC outperforms other methods in all cases, whereas GLEN-GCN outperforms COLES-GCN, COLES-GCN (Stiefel) and contrastive GCN on all datasets.

## 6.3 Comparison of Surrogates of Rank

Table 5 above shows results on four additional surrogates of Rank(**S**):

- Nuclear norm: $R_{\text{NN}}(\mathbf{S}) = \sum_i \sigma_i(\mathbf{S})$.

- $\gamma$-nuclear norm [16]: $R_{\gamma\text{-NN}} = \sum_i \frac{(1+\gamma)\sigma_i(\mathbf{S})}{\gamma+\sigma_i(\mathbf{S})}$.

- $S_p$ norm [28]: $R_{S_p} = \sum_i \sigma_i(\mathbf{S})^p$.

- Geman norm [8]: $R_{\text{Geman}} = \sum_i \frac{\sigma_i(\mathbf{S})}{\gamma+\sigma_i(\mathbf{S})}$.

## 6.4 Transductive One-shot Learning on Image Classification Datasets

The most common setting in FSL is the inductive setting. In such a scenario, only samples in the support set can be used to fine-tune the model or learn a function for the inference of query labels. In contrast, in the transductive scenario, the model has access to all the query data (unlabeled) that needs to be classified.

EASE [54] is a transductive few-shot learner for so-called episodic image classification. Given feature matrix $\mathbf{Z} \in \mathbb{R}^{n \times m}$ from a CNN backbone (ResNet-12), EASE minimizes $\mathrm{Tr}(\mathbf{U}\mathbf{Z}^\top \mathbf{L}_w \mathbf{Z}\mathbf{U}^\top) - \mathrm{Tr}(\mathbf{U}\mathbf{Z}^\top \mathbf{L}_t \mathbf{Z}\mathbf{U}^\top)$ (subject to $\mathbf{U}\mathbf{U}^\top = \mathbf{I}$) in order to learn a linear projection $\mathbf{U}$.

We extend GLEN to EASE to learn the linear projection $\mathbf{U}$ by minimizing $\log \det(\mathbf{U}\mathbf{Z}^\top \mathbf{L}_w \mathbf{Z}\mathbf{U}^\top) - \log \det(\mathbf{U}\mathbf{Z}^\top \mathbf{L}_t \mathbf{Z}\mathbf{U}^\top)$ (subject to $\mathbf{U}\mathbf{U}^\top = \mathbf{I}$. We also apply the $S_p$ norm instead of $\log \det$. Table 6 shows the results of EASE based on the LogDet and the $S_p$-norm based relaxations of GLEN. For the simplicity of experiment, we use soft k-means rather than Sinkhorn k-means as in the EASE pipeline. Please refer to EASE [54] for the experimental setup of one-shot learning.

We evaluate our approach on four few-shot classification benchmarks, mini-ImageNet [38], tiered-ImageNet [32], CUB [41], and CIFAR-FS [21]. The performance numbers are given as accuracy % and the 0.95 confidence intervals are reported. We use publicly available pre-trained ResNet-12 [29] that are trained on the base class training set.

Table 6: Few-shot learning in the transductive setting on EASE based on GLEN.

| methods | *mini*–Imagenet [38] | *tiered*–Imagenet [32] | CIFAR–FS [21] | CUB [41] |
|---|---|---|---|---|
| EASE [54] | 58.2±0.19 | 70.9±0.21 | 65.2±0.21 | 77.7±0.19 |
| EASE GLEN ($S_p$ norm) | 60.5±0.23 | 74.8±0.25 | 67.8±0.25 | 81.5±0.25 |
| EASE GLEN (LogDet) | 61.4±0.23 | 76.4±0.25 | 69.2±0.25 | 83.4±0.25 |

**Scalability.** GraphSAGE and DGI require neighbor sampling with redundant forward/backward steps (long runtime). In contrast, GLEN-S$^2$GC enjoys a simple implementation with low memory usage/low runtime. For graphs with over 100 thousands nodes and 10 millions edges (Reddit), GLEN runs fast on NVIDIA 1080 GPU. Even on larger graph benchmarks, GLEN is fast as it optimizes the total scatter and the within-class matrices whose size depends on embedding size rather than the node number. The runtime of GLEN-S$^2$GC is also favourable in comparison to multi-view augmentation-based GraphCL. Specifically, GLEN-S$^2$GC took 0.54s, 0.3s, 5.3s and 15.4s on Cora, Citeseer, Pubmed and Cora Full, respectively. GraphCL took 110.19s, 101.0s, $\geq$ 8h and $\geq$ 8h respectively. Although the LogDet difference is somewhat slower than the trace difference in forward/backward propagation, it converges faster, thus enjoying a similar low runtime.

## 7 Conclusions

In this paper, we model contrastvie learning as a rank difference problem to approximate the condition that the rank of total scatter matrix should equal the sum of ranks of within-scatter and between-scatter matrices. We relax this NP-hard assumption with a differentiable difference of LogDet terms. We also show two perspectives on GLEN and the existing methods based on the low-rank optimization and distance between symmetric positive (semi-)definite matrices matrices. In low-rank optimization, we explain why the LogDet difference is a better surrogate function to optimize rank difference compared to the trace difference. We also show that our solution encourages linear kernel of embeddings become the geometric mean between the total scatter matrix and the within-class matrix. GLEN works well with many backbones outperforming many unsupervised, contrastive and (semi-)supervised methods.

## Acknowledgments and Disclosure of Funding

We thank reviewers for stimulating questions that helped us improve several aspects of our analysis. Hao Zhu is supported by an Australian Government Research Training Program (RTP) Scholarship. Piotr Koniusz is supported by CSIRO's Machine Learning and Artificial Intelligence Future Science Platform (MLAI FSP).

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
