# Generalized Laplacian Eigenmaps
# (Supplementary Material)

**Hao Zhu**[†,§]     **Piotr Koniusz** [*,§,†]
[§]Data61/CSIRO   [†]Australian National University
allenhaozhu@gmail.com, piotr.koniusz@data61.csiro.au

## A   Spectrum Analysis

Below we analyze the spectrum of embedding matrix $f_{\Theta}(\mathbf{X})$ for several different objectives to show that GLEN enjoys better embedding properties than COLES [63].

Without decorrelation term $\|f_{\Theta}(\mathbf{X})^{\top} f_{\Theta}(\mathbf{X}) - \mathbf{I}\|_F^2$ used by COLES, the trace-based objective in Eq. 6 optimization may results in the so-called dimensional collapse which means that the vast majority of singular values of embeddings are zeroed and only one or two leading components are non-zero.

Fig. 2b shows that in COLES the sixth singular value and the following singular values are almost equal zero, whereas the first singular value is over 2600 in value, thus dominating the spectrum. Fig. 2a shows that our GLEN based on the LogDet difference of terms performs better than COLES. Firstly, the first leading singular value is not as large as COLES (only 80) so it does not dominate the spectrum. The next singular values follow a gradual decline. It is worth noting that even with the decorrelation term $\|f_{\Theta}(\mathbf{X})^{\top} f_{\Theta}(\mathbf{X}) - \mathbf{I}\|_F^2$ used by COLES, it only helps the first five leading singular values, and the first singular value is still larger than the counterpart of GLEN (393 *vs*. 80).

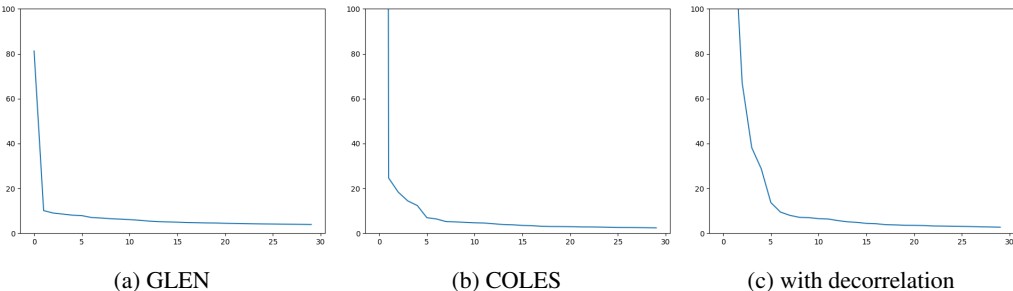

|     (a) GLEN      |      (b) COLES      |    (c) with decorrelation    |

Figure 2: Singular values of embeddings showing the values of first 30 leading singular values. For comparison purposes, we truncate the range of singular values for COLES. The first singular value of COLES is 2650 (very large value). The range of singular values with the use of decorrelation term is 393. The decorrelation term penalizes $\|f_{\Theta}(\mathbf{X})^{\top} f_{\Theta}(\mathbf{X}) - \mathbf{I}\|_F^2$. Notice that the spectrum of GLEN is more balanced.

## B   Nuclear Norm *vs*. LogDet for Rank Minimization

In our paper, to facilitate the analysis we assume $\mathrm{Rank}(\mathbf{S}) \leq \log\det(\mathbf{I} + \mathbf{S}) \leq \mathrm{Tr}(\mathbf{S})$ (kindly note that we have a typo in line 193, where we should have written $\log\det(\mathbf{I} + \mathbf{S})$ rather than $\log\det(\mathbf{S})$). Below we clarify the necessary conditions for this inequality to hold true.

**Proposition 6.** *Let* $\mathbf{S} \in \mathbb{S}_{+(+)}^m$, *if* $\mathrm{Tr}(\mathbf{S}) \geq m$ *then* $\mathrm{Rank}(\mathbf{S}) \leq \log\det(\mathbf{I} + \mathbf{S}) \leq \mathrm{Tr}(\mathbf{S})$.

---

[*]The corresponding author.     Code: `https://github.com/allenhaozhu/GLEN`.

36th Conference on Neural Information Processing Systems (NeurIPS 2022).

*Proof.* It is easy to observe that $\text{Rank}(\mathbf{S}) \leq \text{Tr}(\mathbf{S})$ if $\text{Tr}(\mathbf{S}) \geq m$. Moreover, we have $\text{Tr}(\mathbf{S}) > \text{Tr}(\log(\mathbf{I} + \mathbf{S})) = \log\det(\mathbf{I} + \mathbf{S})$. Let us assume $\text{Rank}(\mathbf{S}) = n \leq m \leq \text{Tr}(\mathbf{S})$, then we have $\log(1 + \text{Tr}(\mathbf{S})) \leq \sum_i^m \log(1 + \sigma_i(\mathbf{S})) \leq \sum_i^m \log(\mathbf{I} + \text{Tr}(\mathbf{S})/m)$ and $\sum_i^m \log(1 + \text{Tr}(\mathbf{S})/m) \geq \sum_i^n \log(1 + m/n) \geq \sum_i^m \log(1 + m/m) \geq \text{Rank}(\mathbf{S})$. $\square$

## C   Gradient Analysis

**Proposition 7.** *The objective of optimization $f_{\boldsymbol{\Theta}}(\mathbf{X})^\top \mathbf{L}_t = f_{\boldsymbol{\Theta}}(\mathbf{X})^\top \mathbf{L}_w$ is achieved when the following gradient rule is zeroed.*

*Proof.*

$$
\frac{\partial \log\det\left(\mathbf{I} + \alpha f_{\boldsymbol{\Theta}}(\mathbf{X})^\top \mathbf{L}_t f_{\boldsymbol{\Theta}}(\mathbf{X})\right)}{\partial f_{\boldsymbol{\Theta}}(\mathbf{X})} = \alpha \left(\mathbf{I} + \alpha f_{\boldsymbol{\Theta}}(\mathbf{X})^\top \mathbf{L}_t f_{\boldsymbol{\Theta}}(\mathbf{X})\right)^{-1} f_{\boldsymbol{\Theta}}(\mathbf{X})^\top \mathbf{L}_t,
$$
$$
\frac{\partial \log\det\left(\mathbf{I} + \alpha f_{\boldsymbol{\Theta}}(\mathbf{X})^\top \mathbf{L}_w f_{\boldsymbol{\Theta}}(\mathbf{X})\right)}{\partial f_{\boldsymbol{\Theta}}(\mathbf{X})} = \alpha \left(\mathbf{I} + \alpha f_{\boldsymbol{\Theta}}(\mathbf{X})^\top \mathbf{L}_w f_{\boldsymbol{\Theta}}(\mathbf{X})\right)^{-1} f_{\boldsymbol{\Theta}}(\mathbf{X})^\top \mathbf{L}_w,
$$
(18)

$$
\frac{\partial \log\det\left(\mathbf{I} + \alpha f_{\boldsymbol{\Theta}}(\mathbf{X})^\top \mathbf{L}_t f_{\boldsymbol{\Theta}}(\mathbf{X})\right)}{\partial \mathbf{Z}} - \frac{\partial \log\det\left(\mathbf{I} + \alpha f_{\boldsymbol{\Theta}}(\mathbf{X})^\top \mathbf{L}_w f_{\boldsymbol{\Theta}}(\mathbf{X})\right)}{\partial \mathbf{Z}} = 0 \Leftrightarrow
$$
$$
\left(\mathbf{I} + \alpha f_{\boldsymbol{\Theta}}(\mathbf{X})^\top \mathbf{L}_t f_{\boldsymbol{\Theta}}(\mathbf{X})\right)^{-1} f_{\boldsymbol{\Theta}}(\mathbf{X})^\top \mathbf{L}_t - \left(\mathbf{I} + \alpha f_{\boldsymbol{\Theta}}(\mathbf{X})^\top \mathbf{L}_w f_{\boldsymbol{\Theta}}(\mathbf{X})\right)^{-1} f_{\boldsymbol{\Theta}}(\mathbf{X})^\top \mathbf{L}_w = 0.
$$
(19)
$\square$

## D   Reproducibility

Table 7: The statistics of datasets.

| Dataset | Task | Nodes | Edges | Features | Classes |
|---|---|---|---|---|---|
| Cora | Transductive | 2,708 | 5,429 | 1,433 | 7 |
| Citeseer | Transductive | 3,327 | 4,732 | 3,703 | 6 |
| Pubmed | Transductive | 19,717 | 44,338 | 500 | 3 |
| Cora Full | Transductive | 19,793 | 65,311 | 8,710 | 70 |
| Ogbn-arxiv | Transductive | 169,343 | 1,166,243 | 128 | 40 |
| Reddit | Inductive | 232,965 | 11,606,919 | 602 | 41 |

### D.1   Datasets

In this paper, we evaluate our method using six datasets.

**Cora** is a well-known citation network that is labelled by paper topic. The majority of approaches report on a subset of this dataset. The **Cora** dataset contains 2708 scientific publications divided into seven categories. The citation network has 5429 links. Each publication in the dataset is described by a 0/1-valued word vector indicating the absence/presence of the corresponding dictionary word. The dictionary contains 1433 distinct words. **Cora Full** is made up of 19793 scientific publications divided into seventy categories. The citation network has 65311 links. The dictionary contains 1433 distinct words.

**CiteSeer** contains 3312 scientific papers divided into six categories. The citation network has 4732 linkages. Each publication in the dataset is described by a 0/1-valued word vector indicating the absence/presence of the relevant dictionary word. The dictionary has 3703 distinct terms.

**Pubmed** is comprised of 19717 scientific papers on diabetes from the PubMed database, grouped into one of three categories. The citation network has 44338 linkages. Each publication in the dataset is described by a TF/IDF weighted word vector drawn from a vocabulary of 500 distinct terms.

**Reddit** is a graph dataset including Reddit postings from September of 2014. In this situation, the node label represents the community, or "subreddit," to which a post belongs. The 50 largest

communities were sampled in order to create a post-to-post graph, which connects posts if the same user comments on both. This dataset has a total of 232,965 posts with an average degree of 492. The first 20 days are used to training, with the remaining days used to testing (with 30% percent used for validation).

**Ogbn-arxiv** is the citation network represented by a directed graph of all Computer Science (CS) arXiv papers indexed by MAG [62]. Each node represents an arXiv paper, and each directed edge indicates that one paper references another. Each paper includes a 128-dimensional feature vector generated by averaging the embeddings of words in the title and abstract. Individual word embeddings are produced by running the skip-gram model [61] over the MAG corpus.

# E   Implementation Details

GLEN is implemented in PyTorch.The propagation procedure is efficiently implemented with sparse-dense matrix multiplications. The codes of GCN, COLES-GCN, SGC, COLES-SGC, $S^2GC$ and COLES-$S^2$GC are also implemented with PyTorch. The weight matrices of classifier are initialized with Glorot normal initializer. We employ Adam [59] to optimize parameters of the proposed methods and adopt early stopping to control the training epochs based on validation loss. For all experiments, we use Adam to optimize Eq. 5 because our method does not have a closed-form solution as COLES does. All the experiments in this paper are conducted on a single NVIDIA GeForce RTX 1080 with 8 GB memory on a PC with Unbuntu 18.04. We use Python 3.7.3, PyTorch 1.10.0, and CUDA 10.2.

## E.1   Hyperparameters

Tables 8 and 9 summarize hyperparameters used by us. GLEN is not highly sensitive to different hyperparameters. The only hyperparameter for SGC and S $^2$GC is the aggregation step $K$. As a result, we use $K = 8$ for the majority of benchmarks. Except for Ogbn-arxiv, all contrastive-based approaches use logistic regression as the classifier. It is worth noting that we do not modify any logistic regression parameters and instead use the default settings. Ogbn-arxiv exhibits non-linear feature characteristics because Bag-of-Words was used by the authors to form embeddings. As a result, the MLP classifier is chosen for GLEN-$S^2$GC on this dataset. We specifically preserve the MLP classifier configuration from the baseline. The learning rate for the MLP is 0.005 and the dropout rate is 0.4.

Table 8: The hyperparameters of datasets (node classification).

| Dataset | Optimizer | K | lr | weight decay | Epoch | hidden size |
|---------|-----------|---|------|--------------|-------|-------------|
| Cora | Adam | 8 | 1e-3 | 5e-4 | 20 | 512 |
| Citeseer | Adam | 8 | 1e-4 | 1e-4 | 80 | 512 |
| Pubmed | Adam | 8 | 2e-2 | 1e-5 | 40 | 256 |
| Cora Full | Adam | 2 | 1e-2 | 0 | 30 | 512 |
| Ogbn-arxiv | Adam | 10 | 1e-2 | 0 | 500 | 126 |
| Reddit | Adam | 2 | 1e-2 | 0 | 100 | 600 |

Table 9: The hyperparameters of datasets (node clustering).

| Dataset | Optimizer | K | lr | weight decay | Epoch | hidden |
|---------|-----------|---|------|--------------|-------|--------|
| Cora | Adam | 8 | 1e-2 | 5e-4 | 1 | 512 |
| Citeseer | Adam | 8 | 1e-4 | 1e-4 | 30 | 512 |
| Pubmed | Adam | 8 | 2e-2 | 1e-5 | 40 | 256 |

# F   Limitation and Broader Impact

Although we propose a promising criterion to learn discriminant features, the rank difference problem is NP-hard. Choosing and especially solving high quality surrogates to the rank difference of terms is still an unsolved optimization problem that requires further work within optimization community.

This paper discusses the design of feature space from a critical point of view. To this end, it proposes a novel generalized graph embedding framework which is very different compared to the traditional graph embedding frameworks (*e.g*., Laplacian Eigenmaps, Linear Discriminant Analysis or Contrastive Laplacian Eigenmaps). The scope of this framework is not limited to graph embedding. GLEN can help many downstream works which require learning discriminant representation in an unsupervised manner to improve the performance of downstream tasks. Node classification, object category recognition, metric learning and other downstream tasks can accelerate downstream training thanks to GLEN. More importantly, GLEN-S$^2$GC is a lightweight network which is energy friendly compared to heavy GCN-based GCL (see 'Scalability' section).

Regarding negative impacts, we are not aware of any. GCL is an unsupervised learning paradigm which, by itself, cannot lead to any negative impact.

# G    Additional Results

**Comparison with other frameworks.**    Below we show how we can redefine Local Preserving Projection (LPP) [58] and Deep Spectral Clustering (DSC) [60] within the COLES and GLEN frameworks. All models below are based on the S$^2$GC backbone:

- DSC extends 'Deep Spectral Clustering Learning' [60], by minimizing $\mathrm{Tr}(f_\Theta(\mathbf{X})^\top \mathbf{L}_w f_\Theta(\mathbf{X}))$ where $f_\Theta$ is a two-layer neural network (MLP). Kindly note this is non-contrastive learning that only uses $\mathbf{L}_w$.
- LPP is extension of 'Locality Preserving Projections' [58], that learns an orthogonal linear projection by minimizing $\mathrm{Tr}(\mathbf{U}\mathbf{X}^\top \mathbf{L}_w \mathbf{X}\mathbf{U}^\top))$. Notice this is non-contrastive learning that only uses $\mathbf{L}_w$.
- COLES-LPP is defined by us as minimizing $\mathrm{Tr}(\mathbf{U}\mathbf{X}^\top \mathbf{L}_t \mathbf{X}\mathbf{U}^\top)) - \mathrm{Tr}(\mathbf{U}\mathbf{X}^\top \mathbf{L}_w \mathbf{X}\mathbf{U}^\top))$.
- COLES-DSC minimizes $\mathrm{Tr}(f_\Theta(\mathbf{X})^\top \mathbf{L}_t f_\Theta(\mathbf{X})) - \mathrm{Tr}(f_\Theta(\mathbf{X})^\top \mathbf{L}_w f_\Theta(\mathbf{X}))$. Two MLP layers are added (as in the DSC model above) between the S$^2$GC backbone and the loss. Their layer parameters are added to $\Theta$.
- GLEN-LPP is defined as $\log\det(\mathbf{U}\mathbf{X}^\top \mathbf{L}_t \mathbf{X}\mathbf{U}^\top)) - \log\det(\mathbf{U}\mathbf{X}^\top \mathbf{L}_w \mathbf{X}\mathbf{U}^\top))$.
- GLEN $\log\det(f_\Theta(\mathbf{X})^\top \mathbf{L}_t f_\Theta(\mathbf{X})) - \log\det(f_\Theta(\mathbf{X})^\top \mathbf{L}_w f_\Theta(\mathbf{X}))$ also uses two MLP layers as in the DSC model.

Table 10 below shows the results.

Table 10: Mean classification accuracy (%) and the standard dev. over 50 random splits. Numbers of labeled samples per class are in parentheses. The best accuracy per column is in bold. Models are organized into semi-supervised, contrastive and unsupervised groups. OOM means out of memory.

| Method | Cora | | Citeseer | | Pubmed | | Cora Full | |
|---|---|---|---|---|---|---|---|---|
| | (5) | (20) | (5) | (20) | (5) | (20) | (5) | (20) |
| S$^2$GC | 71.4±4.4 | 81.3±1.2 | 60.3±4.0 | 69.5±1.2 | 67.6±4.2 | 73.3±2.0 | 41.8±1.7 | 60.0±0.5 |
| LPP | 34.5±1.6 | 54.4±1.5 | 30.5±1.4 | 42.3±1.5 | 39.4±5.3 | 43.9±4.7 | 50.8±1.4 | 61.8±0.5 |
| DSC | 32.5±3.9 | 53.4±4.6 | 37.2±4.0 | 48.2±3.0 | 40.0±5.6 | 39.2±5.6 | 50.0±0.0 | 60.0±1.0 |
| COLES-LPP | 75.0±3.4 | 81.0±1.3 | 67.9±2.3 | 71.7±0.9 | 62.6±5.0 | 73.2±2.6 | 47.6±1.2 | 59.2±0.5 |
| COLES-DSC | 73.7±3.0 | 80.4±1.0 | 67.4±2.0 | 71.9±0.9 | 60.3±6.0 | 65.9±1.7 | 23.0±1.4 | 38.3±1.1 |
| GLEN-LPP | 75.3±3.6 | 82.6±1.2 | 65.9±2.7 | 71.5±1.0 | 68.9±3.9 | 78.4±2.1 | 51.4±1.4 | 62.0±0.6 |
| GLEN-DSC | 78.2±2.4 | 83.0±1.0 | 69.1±2.1 | 72.3±0.9 | 70.6±3.9 | 80.1±1.9 | 53.0±1.5 | 62.6±0.5 |

# H    LogDet model *vs*. JBLD

**Definition 1.** *Given an matrix* $\mathbf{A}, \mathbf{B} \in \mathbb{S}^m_{+(+)}$, *we define the regularized Jensen Bregman LogDet Divergence as follows:*

$$J(\mathbf{A}, \mathbf{B}; \alpha) = \log\det\left(\frac{\mathbf{I} + \alpha\mathbf{A} + \alpha\mathbf{B}}{2}\right) - \frac{1}{2}\big(\log\det(\mathbf{I} + \alpha\mathbf{A}) + \log\det(\mathbf{I} + \alpha\mathbf{B})\big) \ \textit{for} \ \mathbf{A}, \mathbf{B}, \alpha \geq 0,$$

$$(20)$$

*where* $\mathbf{I}$ *can be interpreted as a regularization to prevent* $\det(\mathbf{A}) = 0$ *and* $\log \det(\mathbf{A}) = -\infty$.

**Proposition 8.** *For any* $\mathbf{S}_w, \mathbf{S}_t \in \mathbb{S}_{+(+)}^m$ *and* $\alpha \geq 0$, *we have:*

$$\frac{1}{2}\delta(\mathbf{S}_t, \mathbf{S}_w; \alpha, 1) \geq J(\mathbf{S}_w, \mathbf{S}_b; \alpha)$$

$$= \log \det \left( \frac{\mathbf{I} + \alpha\mathbf{S}_w + \alpha\mathbf{S}_b}{2} \right) - \frac{1}{2} \left( \log \det(\mathbf{I} + \alpha\mathbf{S}_w) + \log \det(\mathbf{I} + \alpha\mathbf{S}_b) \right),$$
(21)

*where the equality holds only if* $\alpha = 0$ *and then* $\det(\mathbf{I} + \alpha\mathbf{S}_b) = 1$.

*Proof.* Based on Eq. 20, we notice JBLD is a lower bound of our loss:

$$J(\mathbf{S}_w, \mathbf{S}_b; \alpha) = \log \det \left( \frac{\mathbf{I} + \alpha\mathbf{S}_w + \alpha\mathbf{S}_b}{2} \right) - \frac{1}{2} \left( \log \det(\mathbf{I} + \alpha\mathbf{S}_w) + \log \det(\mathbf{I} + \alpha\mathbf{S}_b) \right)$$

$$= m \log 0.5 \cdot \log \det (\mathbf{I} + \alpha\mathbf{S}_t) - \frac{1}{2} \left( \log \det(\mathbf{I} + \alpha\mathbf{S}_w) + \log \det(\mathbf{I} + \alpha\mathbf{S}_b) \right) \quad (22)$$

$$\leq \frac{1}{2} (\log \det (\mathbf{I} + \alpha\mathbf{S}_t) - \log \det(\mathbf{I} + \alpha\mathbf{S}_w)) = \frac{1}{2}\delta(\mathbf{S}_t, \mathbf{S}_w; \alpha, 1).$$

$\square$

## I   Derivation of GLEN from the SampledNCE framework

COLES [63] extends Laplacian Eigenmaps:

$$\underset{\mathbf{Z}, \; s.t. \Omega(\mathbf{Z})}{\arg\min} \; W_{ij}^+ \|\|\mathbf{z}_i - \mathbf{z}_j\|\|_2^2, \tag{23}$$

where $\mathbf{Z} = [\mathbf{z}_1, \cdots, \mathbf{z}_n]$ and $\Omega(\mathbf{Z})$ are constraints (*i.e.*, orthogonality) by expanding the SampledNCE formulation:

$$\mathbb{E}_{i \sim p_d} \left[ \mathbb{E}_{j \sim p_d(j|i)}[s_\Theta(x_i, x_j)] + \eta \, \mathbb{E}_{j' \sim p_n(j'|i)}[\tilde{s}_\Theta(x_i, x_{j'})] \right]. \tag{24}$$

Symbols $p_n$ and $p_d$ are negative and positive sampling distributions respectively, $s_\Theta(v, u) = \log \exp(\mathbf{u}^\top \mathbf{v}) = \mathbf{u}^\top \mathbf{v}$ and $\tilde{s}_\Theta(v, u') = \log \exp(-\mathbf{u}'^\top \mathbf{v}) = -\mathbf{u}'^\top \mathbf{v}$ are similarity measures, whereas $\eta \geq 0$ controls the impact of negative sampling.

GLEN generalizes SampledNCE, a framework for contrastive learning with positive and negative sampling, which relies on two terms:

$$\mathbb{E}_{v \sim p_d(v)} \left[ \mathbb{E}_{u \sim p_d(u|v)} s(\mathbf{u}, \mathbf{v}) \right] \quad \text{and} \quad \eta \, \mathbb{E}_{v \sim p_d(v)} \left[ \mathbb{E}_{u' \sim p_n(u'|v)} \tilde{s}(\mathbf{u}', \mathbf{v}) \right]. \tag{25}$$

The above two terms are evaluated over two different distributions $u \sim p_d(u \mid v)$ (nodes $u$ from the adjacency matrix) and $u' \sim p_n(u' \mid v)$ (nodes $u'$ from random negative adjacency matrix).

Let us consider the positive sampling term (the negative sampling term can be expanded in the similar way). Let $p_d(v) = \frac{1}{\sqrt{D_{vv}}}$ and $p_d(u \mid v) = \frac{\hat{W}_{uv}}{\sqrt{D_{uu}}}$ where $\hat{\mathbf{W}}$ is an unnormalized adjacency matrix and $\mathbf{D}$ is its degree matrix. Let $\mathbf{W}$ be the degree normalized matrix. Notice $u$ and $v$ are indexes of embeddings $\mathbf{u}$ and $\mathbf{v}$. Let $s(\cdot)$ be defined as in COLES. Then:

$$\mathbb{E}_{v \sim p_d(v)}[\mathbb{E}_{u \sim p_d(u|v)}, s(\mathbf{u}, \mathbf{v})] = \sum_{u,v} W_{uv}, s(\mathbf{u}, \mathbf{v}) = \sum_{i=1}^m \sum_{u,v} W_{uv} u_i v_i = \phi(\mathbf{Z}^\top \mathbf{W} \mathbf{Z}), \tag{26}$$

where $\phi(\cdot)$ is a pooling function, *i.e.*, $\phi(\mathbf{M}) = \mathrm{Tr}(\mathbf{M})$ yields COLES:

$$\sum_{i=1}^m \sum_{j=1}^m \delta(i-j)\phi(\mathbf{z}_i^\top \mathbf{W} \mathbf{z}_j) = \sum_{i=1}^m \phi(\mathbf{z}_i^\top \mathbf{W} \mathbf{z}_i) \quad \text{if} \quad \mathbf{z}_i \perp \mathbf{z}_j \text{ for } i \neq j, \tag{27}$$

where $\mathbf{z}_i \perp \mathbf{z}_j$ imposes orthogonality constraints of Laplacian eigenmaps and $\delta(z) = 1$ if $z = 0$ and $\delta(z) = 0$ if $z \neq 0$. Finally, think that rows of $\mathbf{Z}$ contain all $\mathbf{u}$ (and $\mathbf{v}$).

We let the pooling operator $\phi(\cdot)$ operate on the entire spectrum under a general aggregation scheme. A very general operator is $\phi(\mathbf{M}) = \text{Rank}(\mathbf{M})$ from which we can recover the original trace (nuclear norm) of COLES or LogDet of GLEN, $\gamma$-nuclear, $S_p$, and Geman norm, respectively.

COLES uses the following expression based on SampledNCE:

$$\min_{\mathbf{Z}} \sum_{i,j} W_{ij}^+ \|\mathbf{z}_i - \mathbf{z}_j\|_2^2 - \left( \frac{\eta}{\kappa} \sum_{l=1}^{\kappa} W_{ij}^{l,-} \right) \|\mathbf{z}_i - \mathbf{z}_j\|_2^2 = \max_{\mathbf{Z}} \text{Tr}(\mathbf{Z}^\top \mathbf{L}_t \mathbf{Z}) - \text{Tr}(\mathbf{Z}^\top \mathbf{L}_w \mathbf{Z}), \quad (28)$$

where $\mathbf{W}^+$ is a normalized adjacent matrix and $\mathbf{W}^{l,-}$ are $\kappa$ normalized randomized $k$-regular graphs (adjacency matrices), while $\mathbf{L}_w$ and $\mathbf{L}_t$ are the corresponding Laplacian matrices.

Negative random sampling is represented by $\mathbf{W}^-$, *e.g.*, a randomized $k$-regular graph or several such graphs.

If we sample $\kappa \to \infty$ randomized $k$-regular graphs (adjacency matrices of size $n \times n$) (each row receives 1 with probability $k/n$), the expectation of randomized graph (adjacency matrix) is given as:

$$\mathbb{E}[\mathbf{W}^-] = \lim_{\kappa \to \infty} \frac{1}{\kappa} \sum_{l=1}^{\kappa} \mathbf{W}^{l,-} = \frac{k}{n} \mathbf{1}\mathbf{1}^\top, \quad (29)$$

which by itself is a fully-connected graph with the graph Laplacian matrix $\mathbf{L}_t = \mathbf{I} - \frac{k}{n} \mathbf{1}\mathbf{1}^\top$.

We simply set $k = 1$ to use 1-regular graphs for negative sampling so $\mathbf{L}_t = \mathbf{I} - \frac{1}{n} \mathbf{1}\mathbf{1}^\top$. Thus, our contrastive term is equivalent of the total scatter matrix $\mathbf{S}_t$ known from the Linear Discriminant Analysis, *i.e.*, $\mathbf{S}_t = \mathbf{Z}^\top (\mathbf{I} - \frac{1}{n} \mathbf{1}\mathbf{1}^\top) \mathbf{Z} = \mathbf{Z}^\top \mathbf{L}_t \mathbf{Z}$. The positive sampling is encoded by the graph adjacency matrix $\mathbf{L}_w$.

Thus, our GLEN is given as:

$$\arg\max_{\mathbf{Z}} \ \text{Rank}(\mathbf{Z}^\top \mathbf{L}_t \mathbf{Z}) - \text{Rank}(\mathbf{Z}^\top \mathbf{L}_w \mathbf{Z}). \quad (30)$$

More precisely, if $\mathbf{Z}$ is produced by an encoder $f_\Theta(\mathbf{X})$ then we optimize the above problem w.r.t. $\Theta$.