# OpenReview forum: "Generalized Laplacian Eigenmaps"
_NeurIPS.cc/2022/Conference — NeurIPS 2022 Accept_

### Official Review · Reviewer_3VJW · 2022-06-17

**Rating:** 5
**Confidence:** 4
**Soundness:** 2 fair
**Presentation:** 2 fair
**Contribution:** 3 good

**Summary:**

The paper proposes a novel objective for graph embedding, called Generalized Laplacian EigeNmaps (GLEN), to learn graph representation by maximizing the difference of logdet between the total scatter matrix and the within-class scatter matrix. The authors interpret this as a surrogate of rank difference maximization and give some theoretical results. Experiments show that GLEN offers good accuracy and scalability against state-of-the-art baselines on various benchmarks.

** Post Rebuttal Update **

I've read the rebuttal and the other reviewers' comments. I appreciate the update the authors have made, for example, the (supposedly) new experimental updates regarding Reviewer bffF's comment. I appreciate the experimental results against the prior art. My general concern is whether the theorem indeed shows a difference from the prior art. It seems to me that the rank difference formulation or the minimum class separation has been identified in the literature. What's more interesting is to explain why the logdet can be a better objective, which seems quite possible given the new experimental results.




**Questions:**

Q1. The motivation of Condition 1 is unclear. In particular, why $\text{Rank}(S_t)=\text{Rank}(S_w)+\text{Rank}(S_b)$ yields good embedding?

Q2. Why formulate the main problem as rank difference? Why not directly analyze the logdet difference?



**Limitations:**

The authors didn't discuss the limitation and potential social impact.

**Strengths And Weaknesses:**

** Strengths **

S1. The proposed algorithm is based on the scatter matrices, which are of the size of $d\times d$, not $n\times n$. Note that $d$ and $n$ are the embedding dimension and the node number, respectively. Thus, the method is quite scalable to large graphs.

S2. Experiments show strong results in various settings and datasets.

** Weaknesses **

W1. The authors approximate the rank difference with logdet difference. However, it is unclear why optimizing the rank difference or the logdet difference leads to good results.

W2. The notations are confusing and may contain errors. For example, in section 3.1, it seems that $Z$ is $d\times n$ instead of $n\times m$. Also, the $S$ matrices should be $d\times d$ instead of $n\times n$. If $Z$ is $n\times m$ and $S$ is $n\times n$, then the algorithm should not be scalable as $n$ is the number of nodes.

---

> ### Author Response · Authors · 2022-08-02
> **Response to Rev. 3 (3VJW)**
>
> # Response to Rev. 3 (3VJW)
>
> ***We thank the reviewer** for the constructive review and interesting questions.*
>
> ## 1. The notations are confusing and may contain errors.
> Thank you and we apologize. There are indeed some errors which resulted from rushed editing. We have now carefully revised these notations and will upload revised paper shortly.
>
> Indeed, the  ${\bf S}$ matrices are of size $d\times d$ which is the reason why our algorithm can scale up to large graphs with $n$ nodes.
>
> ## 2. The motivation of Condition 1 is unclear. Why does it yield good embedding?
>
> * Thank you. Condition 1 yields a nice property described in Theorem 1: **the feature space under Condition 1 has the property of minimum separation  between any two embeddings from two different classes (best minimum margin)**, which is equal to the distance between the corresponding class centers. In contrast, Linear Discriminant Analysis (LDA) strives to separate class centres and shrink within-class variance but cannot guarantee anything for pairs of embeddings.
>
>
>
>
> ## 3. Why optimizing the Rank difference or the LogDet difference leads to good results?
>
> Despite Condition 1 provides us nice guarantees on the minimum separation between embeddings from two classes (best minimum margin), it is an NP-hard problem and so its solution needs relaxation.
>
> In Resp. 2 to Rev. 1 (bffF) and Resp. 3 to Rev. 2 (Vcae), **we show that the LogDet difference formulation, in its limit case, can yield the exact Rank difference**. Under another set of parameters, **it can reverse to the Trace difference problem akin to COLES and LDA**. For this reason, **the LogDet difference can be tuned towards the guarantee in Theorem 1**.
>
>
>
> ## 4. Why formulate the main problem as rank difference? Why not directly analyze the LogDet difference?
>
> * **We propose the Rank difference because we need to reformulate the NP-hard problem of Condition 1 as a measurable function that can be optimized**.
> The LogDet difference is a great surrogate from which we can recover either the Trace difference or even the exact Rank difference (in the limit case, of course). In practice, the limit case looses smoothness and cannot be optimized, but the trade-off compromise can.
>
> * We also note that the  Rank difference problem is universal. Its relaxations can yield the Trace difference (COLES, LDA) or LogDet difference (objective used in our GLEN). This facilitates the creation of a unified objective function.
>
> **We truly hope that with help of responses addressing theoretical aspects, we are able to convince reviewer about the value of our work**. We apologize for referencing responses to other reviewers but we felt it makes more sense than repeating them.

---

### Official Review · Reviewer_Vcae · 2022-07-08

**Rating:** 6
**Confidence:** 4
**Soundness:** 3 good
**Presentation:** 3 good
**Contribution:** 2 fair

**Summary:**

This paper proposes a new unsupervised representation learning method, mainly based on GNN.

The idea is motivated by the scatter matrices that are usually used in LDA. Based on the fact that the features would be discriminative provided that Condition 1 holds, the model aims to maximize rank($S_w$) and minimize rank($S_b$) simultaneously, which is different from the losses of the popular contrastive learning.

In Section 3, the authors show that the equivalence between the specific two-layer (featureless) GAE and linear (featureless) GAE.

In Section 4, the authors try to investigate the real impact of ReLU on the hidden layer.

Then, as the original goal is NP-hard, a surrogate that approximates the rank better than the classical nuclear norm is introduced.

Finally, sufficient experiments are conducted to verify the idea.



**Questions:**

(More details can be found in the previous part)

1. Why do you constrain the model on the GNNs? In other words, why not conduct experiments on the general datasets?>

2. Is there a difference between the following literature [1] and this paper? It limits the novelty of the paper.

   [1] Calibrated Multi-Task Learning, SIGKDD, 2018.

3. Could the authors also provide some experiments under the common settings of Cora/Citeseer/PubMed, instead of the random split?

**Ethics Review Area:**

["I don’t know"]

**Strengths And Weaknesses:**

### Pros:

1. The idea to use the scatter matrices to learn discriminative features seems novel. It is different from the popular contrastive models.
2. The motivation is convincing and interesting to me.
3. The experimental results, especially on semi-supervised node classification when labels are pretty rare, seem to show effectiveness.

### Cons:

1. An important question that confuses me is why not to testify the idea on the setting of general contrastive learning.  If I don't misunderstand the model, the graph (*i.e.*, adjacency) seems to be only used in the implementation of $f_\Theta$, which indicates that $f_\Theta$ could be any neural networks (or other projection techniques).  So why do you constrain the model on the GNNs? If some similar ideas have been proposed in the general contrastive learning (which I'm not familiar with the newest publications), it will severely affect the novelty.

2. A major concern is the surrogate may be not novel. The idea to use $\log(\cdot)$ to replace the $\ell_p$-norm (which is equivalent to the Schatten-$p$ norm for the rank) has been well studied. Is there a difference between the following literature and this paper? It limits the novelty of the paper.

   [1] Calibrated Multi-Task Learning, SIGKDD, 2018.

3. Could the authors also provide some experiments under the common settings of Cora/Citeseer/PubMed, instead of the random split? It is also an important comparison with the existing GNN models.

4. No source code is provided so that it may limit the reproducibility.

5. There are some typos including but not limited to:
   - The meaning of letters in boldface is confusing. For example, in Figure 1, the matrix is denoted by $S_b$ while in Section 3.1, all matrices are highlighted by boldface (*e.g.*, $\textbf{S}_w$). In Line 135，$C$ is also bold.
   - In Line-145, Theorem 2 -> Theorem 1?

Overall, I would like to update my score after reading other reviews and the response.

---

> ### Author Response · Authors · 2022-08-02
> **Response to Rev. 2 (Vcae)  (part II of II)**
>
> # Response to Rev. 2 (Vcae)  (part II of II)
>
> ## 3. Major concern is the surrogate may be not novel.
>
> Thank you. Kindly note that we do not claim that LogDet surrogate of Rank minimization is our contribution.
>
> Apart from reasons described in above responses, we choose LogDet  due to its stably during  backpropagation, smoothness and ability to recover several other models (Trace for COLES, or even the exact Rank in the limit). Important is also that **thanks to Condition 1 (main paper) we can strive for the exact separation guarantee between any two embeddings of two different classes (best minimum margin), and we can drop the orthogonality constraint on Laplacian eigenmaps embedding**, as detailed in Resp. 1 to Rev. 1 (bffF).
>
> In fact, even compared with just the LogDet surrogate of Rank from `Calibrated Multi-Task Learning, SIGKDD, 2018', if applied to our Rank difference problem, the Rank difference error between just two matching eigenvalues $\sigma\_i$ and $\sigma'\_i$ of matrices ${\bf A}$ and ${\bf B}$ would be:
> $$
> \Delta\epsilon=\log(\sigma\_i +1)-\log(\sigma'\_i+1),
> $$
> which indicates that for large eigenvaleus and large gap $\|\sigma\_i-\sigma'\_i\|$ , the error is large.
>
> **In our case, in the limit case, our formulation enjoys the exact difference of Ranks** (use $\gamma=1$):
> $$
> \Delta\epsilon=\lim_{\alpha\rightarrow\infty}\frac{1}{\log(\alpha+\gamma)}(\log(\alpha\sigma\_i +\gamma)-\log(\alpha\sigma'\_i+\gamma))=0,
> $$
>
>
>
>
> ## 4.  Provide experiments under the common settings of Cora, Citeseer, PubMed, instead of the random splits.
>
> |                           | Cora  | Citeseer | Pubmed  |
> |---------------------------|-----------|-----------|--------------|
> |DeepWalk+F | 77.36  | 64.30  | 69.65 |
> |Node2vec+F | 75.44 |  63.22 |  70.6 |
> |GAE | 73.68 | 58.21 |  76.16 |
> |VGAE | 77.44 |  59.53 |  78.00 |
> |DGI  | 81.26 | 69.50 |   77.70 |
> |GRACE | 80.46 |  68.72 |  80.67 |
> |GraphCL | 81.89 | 68.40 |  OOM |
> | GMI | 80.28| 65.99| OOM |
> | GLEN-S$^2$GC  | **85.1** |**71.9** | 80.72 |
>
> OOM means out-of-memory error on Nvidia RTX 11GB.
>
> Kindly note that our method does not include any graph augmentations or multi-view learning which are orthogonal and complementary directions to ours. GLEN simply uses the adjacency to capture similar node pars, and the random fully-connected dense graph for negative sampling.
>
> ## 5. No source code is provided so that it may limit the reproducibility.
>
> Thank you, **we will of course  release the full code** in due course. In the supplementary material, **we have updated now a simple demo code** for Cora and provided logs on other larger datasets.
>
> ## 6. Some typos.
> Thank you very much for pointing out these typos. We have now revised them accordingly. We plan to revise the paper and upload to the system within one week.
>
> **We truly hope that the above clarifications are able to convince reviewer about the value of our work.**

---

> ### Author Response · Authors · 2022-08-02
> **Response to Rev. 2 (Vcae)  (part I of II)**
>
> # Response to Rev. 2 (Vcae)  (part I of II)
>
> ***We thank the reviewer** for the constructive review and interesting questions.*
>
> ## 1. Why do you constrain the model on GNNs? Why not conduct experiments on general datasets?
>
> * As shown in Eq.1 and 3, our method depends on an adjacency matrix ${\bf S}_w=f\_{\Theta}({\bf X})^\top{\bf L}\_wf\_{\Theta}({\bf X})$ (not just for a GCN encoder) or some notion of label information (one-hot label vectors can be easily used to form an adjacent matrix) for the within-class scatter matrix. GLEN also requires negative sampling realized by the randomly formed negative graph. Kindly see Resp. 1 to Rev. 1 (bffF) to see how randomized $k$-regular graphs form our negative ${\bf L}_t$.
>
> * In unsupervised representation learning for GNNs, an adjacency matrix is readily available, and it can be utilized according to the SampledNCE framework to form positive node pairs in contrastive learning ${\bf L}\_w$. Thus, GLEN is not a generic contrastive learning framework.
>
> * As $f_\Theta$  can be any neural network (or some projection technique), in our experiment we demonstrate GLEN with S$^2$GC and GCN backbones.
>
> * **Below we include an interesting setting of transductive one-shot learning** (images+CNN backbone) where negative graph is also based on fully-connected graph. **EASE (CVPR'22)** minimizes $\text{Tr}(\mathbf{U}\mathbf{X}^\top\mathbf{L}\_w \mathbf{X}\mathbf{U}^\top)-\text{Tr}(\mathbf{U}\mathbf{X}^\top \mathbf{L}\_t \mathbf{X}\mathbf{U}^\top), \\;\text{s.t.}\\; \mathbf{U}\mathbf{U}^\top=\mathbf{I}$ for learning some linear projection $\mathbf{U}$.
>
>   **We extend GLEN to the EASE pipeline** to learn the linear projection $\mathbf{U}$ by minimizing $\text{LogDet}(\mathbf{U}\mathbf{X}^\top\mathbf{L}\_w \mathbf{X}\mathbf{U}^\top)-\text{LogDet}(\mathbf{U}\mathbf{X}^\top \mathbf{L}\_t \mathbf{X}\mathbf{U}^\top), \\; \text{s.t.}\\; \mathbf{U}\mathbf{U}^\top=\mathbf{I}$ (or we use $S_p$ norm instead of LogDet) and we achieve the following results:
>
>   |  |miniImagenet|tieredImagenet|CIFAR-FS|CUB|
>   |-|-|-|-|-|
>   |EASE (CVPR2022)| 58.2±0.19|70.9±0.21|65.2±0.21|77.7±0.19|
>   |EASE GLEN ($S\_p$ norm)| 60.5±0.23|74.8±0.25|67.8±0.25|81.5±0.25|
>   |**EASE GLEN (LogDet)**|**61.4±0.23**|**76.4±0.25**|**69.2±0.25**|**83.4±0.25**|
>
>   Kindly note for the simplicity of ablation, we have used the soft k-means rather than Sinkhorn k-means in the EASE pipeline. The backbone used is ResNet-12 but we are more than happy to supply more backbones if the reviewer would like that (kindly let us know).
>
>
>
> ## 2. Is there a difference between `Calibrated Multi-Task Learning, SIGKDD, 2018' and GLEN?
>
> Thank you for sharing the above paper (we will cite it accordingly). We sincerely think GLEN is novel. The paper pointed by the reviewer uses an MSE loss combined with LogDet based regularizer for the application of multi-task learning (nothing to do `per se' with the Rank difference problem explored by us or contrastive learning), as detailed below.
>
> * Kindly note that our GLEN is generalizing the SampledNCE framework to the matrix form under general pooling operator $\phi(\cdot)$ as detailed in Resp. 1 to Rev. 1 (bffF). From Condition 1 (main paper), we arrive at **Theorem 1 which gives us a guarantee on the minimum separation of any two embeddings from two different classes (best minimum margin which is our target)**. Our general pooling operator is $\phi({\bf M})=\text{Rank}({\bf M})$ because it can be arbitrarily well approximated by the number of various norms resulting in different models, e.g., the nuclear norm (COLES), LogDet (GLEN), $\gamma$-nuclear norm,  $S\_p$ norm,  Geman norm, etc.
>
> * In Resp. 2 to Rev. 1 (bffF) we also show that the **LogDet formulation can recover the Trace formulation (COLES) or even converge to our proposed Rank difference model** in Eq. 3 (main paper). This is one of reasons why we choose LogDet as a versatile operator. In Resp. 6 to Rev. 1 (bffF) we also explain how, **in the limit, the difference of LogDet operators  yields approximation error $\Delta\epsilon=0$**. We also show how **the LogDet difference is lower- and upper-bounded by the Affine Invariant Riemannian Metric (AIRM)** which relates the LogDet difference to non-Euclidean distances, e.g., AIRM, which are however notoriously numerically unstable/almost intractable when backpropagating through in an end-to-end model as ours.
>
>
> * Kindly note that **the Rank Minimization Problem (RMP) is a well known NP-hard classic problem**. RMP arises in diverse areas such as control, system identification, statistics, signal processing, and computational geometry. In our paper, we cited [8] which proposes LogDet heuristic for RMP. Notice we show in Resp. 2 to Rev. 1 (bffF) how to recover from the LogDet formulation the nuclear norm (trace) and the exact rank.
>
>   > [8] Maryam Fazel, Haitham Hindi, and Stephen P Boyd. Log-det heuristic for matrix rank minimization with applications to hankel and euclidean distance matrices.

---

### Official Review · Reviewer_bffF · 2022-07-11

**Rating:** 5
**Confidence:** 4
**Soundness:** 2 fair
**Presentation:** 3 good
**Contribution:** 3 good

**Summary:**

This paper proposes a novel graph-embedding framework, which is a rank difference model. This rank model is NP-hard to solve, so the authors optimize the loss formulation by means of the logdet. The transformed model is then solvable and proven to be theoretically effective. The paper also offers connection between the given model and other graph embedding methods, and calculate the upper and lower bound of the proposed loss function. In total, the paper is well written and theoretically innovative.

**Questions:**

1. Can authors give the proof for the Proposition 1? This proposition lacks the necessary proof.
2. Are there any other cases except COLES that can be generalized to GLEN? If so, please supply examples of these cases. Otherwise, what’s the meaning of the ‘generalization’ of GLEN?
3. Is the logdet to solve the rank problem the original work of the paper? If not, listing the references is necessary. Does the logdet framework still hold the generalization property？If so, please give the proof.
4. Is there any other effective methods to solve the rank difference problem? The authors should compare these methods and explain the reason why logdet is chosen.


**Limitations:**

The generalization of the proposed framework is not clearly described,  and many other cases need to supply except COLES. Besides, there are other methods to solve the rank difference problem except using logdet. The paper does not list and compare these methods.

**Strengths And Weaknesses:**

Strength
1. The COLES can be the special case for the proposed GLEN framework, and the GLEN outperforms the COLES.
2. The theoretical analysis gives the upper and lower bound of the proposed GLEN, which makes the model have good interpretation.
3. The experiments are effective to demonstrate the proposed framework does have good performance.

Weakness
1. It is not clear about the relationship between the proposed model and the contrastive learning, although the contrastive learning is introduced in related work.
2. Compared with the trace model in COLES, the generalization of the proposed rank difference framework is not illustrated clearly. The author should give proof how the trace model can be generalized into the rank model as a special case.
3. Although GLEN is called the generalized Laplacian Eigenmaps framework, the paper shows no other cases that can also be generalized to GLEN except COLES.
4. The paper does not compare the logdet model with other methods that can serve as a surrogate of the rank problem. There are many methods can be used to solve the problem at present, so the authors should list and compare these methods and explain why the logdet is chosen.
5. The logdet terms are chosen to solve the rank model, which actually transfers the rank model into a logdet model. Does the logdet model still maintain the generalization property? If so, please give the proof.

---

> ### Author Response · Authors · 2022-08-02
> **Response to Rev. 1 (bffF) (part IV of IV)**
>
>
> # Response to Rev. 1 (bffF) (part IV of IV)
>
> ## 5. Compare the LogDet model with other surrogates of the Rank problem.
>
> This is indeed a very interesting evaluation to perform. To this end, we choose four different surrogates of $\text{Rank}(\mathbf{S})$:
> * Nuclear norm $R_{N}(\mathbf{S})=\sum_i \sigma_i(\mathbf{S})$
> * $\gamma$-nuclear norm $R_{\gamma\\,NN}=\sum_i\frac{(1+\gamma)\sigma_i(\mathbf{S})}{γ+\sigma_i(\mathbf{S})}$
> * $S\_p$ norm $R_{S_p\\,norm} = \sum_i\sigma_i(\mathbf{S})^{p}$
> * Geman norm $R_{Geman}=\sum_i\frac{\sigma_i(\mathbf{S})}{γ+\sigma_i(\mathbf{S})}$
>
> Below are results on different specific surrogates:
>
> |                           | Cora (5)  | Cora (20) | Citeseer (5) | Citeseer (20) |  Pubmed (5)  | Pubmed (20) | Cora-full (5) | Cora-full (20) |
> |---------------------------|-----------|-----------|--------------|---------------|-----------|-----------|--------------|---------------|
> | GLEN (nuclear norm)                 | 76.5±2.6  | 81.5±1.2  | 67.5±2.2     | 71.3±1.0      | 66.0±5.2  | 77.4±1.9  | 50.8±1.4     | 61.8±0.5      |
> | GLEN ($\gamma$ nuclear)                     | 68.2±3.2   | 80.9±1.3  | 65.8±2.4     | 70.9±1.0     | 67.6±8.1   | 74.4±3.9  | 49.9±4.1     | 57.0±1.0     |
> |GLEN ($S\_p$ norm)                    | 78.0±2.3   | 82.9±1.1  | 67.4±1.9     | 71.7.±1.0     | 62.0±5.7   |  74.9±2.9   |  49.9±1.5      |  60.0±1.6      |
> | GLEN (Geman norm)                   | 65.8±3.4   | 80.1±1.3  | 64.0±2.8     | 70.6.±1.0     | 57.9±5.0   |  67.5±5.6   |  45.1±3.0      |  57.9±1.4      |
> | GLEN (LogDet)                 | **78.2±2.4** | **83.0±1.0** | **69.1±2.1** | **72.3±0.9** | **70.6±3.9** | **80.1±1.9** | **53.0±1.5** | **62.6±0.5**   |
>
> From the table we can conclude that $S\_p$ norm is an interesting approximation of the rank problem. However, on balance, LogDet has been consistently the best performer.
>
>
>
>
> ## 6. Is the LogDet used to solve the Rank problem the original work of the paper? If not, list the references.
>
> * Thank you. The Rank difference and the LogDet difference emerging from our SampledNCE derivations are original, together with Theorem 1. In Resp. 2 we also show that LogDet can indeed approximate Rank with a desired accuracy. The better the approximation, the closer the guarantees hold. As LogDet is typically upper bounded by the Trace [8], this suggests LogDet is closer to fulfilling Theorem 1 than the Trace problem (and COLES).
>
> * Kindly note that we do not claim that LogDet and its association to the rank approximation are our contributions. To that end, we have cited the paper which approximates the rank by LogDet. See [8]. We will make it clearer.
>
>   >[8]. Maryam Fazel, Haitham Hindi, and Stephen P Boyd. Log-det heuristic for matrix rank mini336 mization with applications to hankel and euclidean distance matrices. In Proceedings of the 337 2003 American Control Conference, 2003., volume 3, pages 2156–2162. IEEE, 2003.
>
> * However, approximating the Rank difference and the LogDet difference is a new problem. To that end we have provided:
>   * Proposition 5 (main paper): it shows that the difference of two LogDet terms is lower-bounded by the identity regularized Affine Invariant Riemannian Metric (AIRM) and upper-bounded by $\sqrt{d}$ times AIRM ($d$ is the side size of square matrix). This indicates the relation of the LogDet difference and well-established AIRM for symmetric positive definite matrices.
>   * Below we also provide a theoretical analysis of the approximation error for the LogDet difference. This analysis is important as LogDet is generally unbounded (no finite $\tau\ll\infty$ if eigenvalues are unbounded), i.e., $\lim_{\sigma\_i\rightarrow\infty}\frac{1}{c}\log(\alpha\sigma\_i+\gamma)\rightarrow\infty$ for $c=\log(\alpha+\gamma)$ and $1<\alpha+\gamma\ll\infty$.   Without the loss of generality, let $\gamma=1$, the smallest error under $\alpha\rightarrow\infty$ is $\Delta\epsilon=\lim_{\alpha\rightarrow\infty}\frac{1}{c}(\log(\alpha\sigma\_i +1)-\log(\alpha\sigma'\_i+1))=0$.
>
>
>
> ## 7. Proof for the Proposition 1
> Thank you. We have indeed missed it in the main paper. We have added it into the revision.
>
> The proof follows from the equality $\det(\mathbf{I} + \alpha\mathbf{X}) = \prod_i\sigma_i(\mathbf{I} + \alpha\mathbf{X}) = \prod_i(1 + \alpha\sigma_i(\mathbf{X})) = \det(\mathbf{I} + \alpha\text{Eig}(\mathbf{X}))$ where $\text{Eig}(\cdot)$ is the diagonal matrix with $\sigma_i,\ldots,\sigma_d$. Thus:
>
> $\delta_{rf}(\mathbf{X},\mathbf{X}';\alpha) = \log\det(\mathbf{I}+\alpha\mathbf{X})-\lambda\log\det(\mathbf{I} + \alpha\mathbf{X}') = $
>
> $\log\det(\mathbf{I}+\alpha\text{Eig}(\mathbf{X}))-\lambda\log\det(\mathbf{I} +\alpha\text{Eig}(\mathbf{X}'))=\delta_{rf}(\text{Eig}(\mathbf{X}),\text{Eig}(\mathbf{X}');\alpha)$.

---

> ### Author Response · Authors · 2022-08-02
> **Response to Rev. 1 (bffF) (part III of IV)**
>
> # Response to Rev. 1 (bffF) (part III of IV)
>
> ## 4.  Are there other cases except COLES that can be generalized to GLEN?
>
> Firstly, allow us highlight our contributions from three different perspectives.
>
> 1. We define a measurable condition $\text{Rank}(S_t)=\text{Rank}(S_w)+\text{Rank}(S_b)$ for a class of embedding spaces. Under this condition, our Theorem 1 (main paper) provides a target for the minimum separation between any two embeddings from two different classes (best minimum margin).
>
> 2. For this condition, we design an unconstrained objective function: we maximize the rank difference between $\text{Rank}(S_t)$ and $\text{Rank}(S_w)$. We choose  the rank difference as a variety of solutions can be recovered from it, e.g., the difference of the nuclear norms (COLES), the difference between LogDet expressions, the difference of $\gamma$-nuclear norms, the difference of $S\_p$ norms, and the difference of Geman norms.
>
>    Our optimization problem applies to Laplacian Eigenmaps, Contrastive Laplacian Eigenmaps, Linear Discriminant Analysis and is a matrix-form generalization of the SampledNCE framework, as explained in Resp. 1 above. Thus, our approach is versatile.
>
> 3. Notice that the rank difference is a difficult NP-hard problem. Inspired by the approximation of rank minimization, we choose LogDet difference as a versatile surrogate of the rank difference as it lets recover the trace difference (COLES) and rank difference (GLEN), depending on parameters, as explained in Resp. 2 above. LogDet is differentiable and can approximate the rank with an arbitrary accuracy.
>
>
> 4. Below we show how we can redefine Local Preserving Projection (LPP) and Deep Spectral Clustering (DSC) within the COLES and GLEN frameworks. All models below are based on the S$^2$GC backbone.
>
>   * DSC is extension of `Deep Spectral Clustering Learning', ICML'17, by minimizing $\text{Tr}(f_\Theta(\mathbf{X})^\top\mathbf{L}\_w f_\Theta(\mathbf{X}))$ where $f_\Theta$ is a two-layer neural network (MLP). Kindly note this is non-contrastive learning that only uses $\mathbf{L}\_w$.
>
>   * LPP is extension of `Locality Preserving Projections', NeurIPS'03, that learns an orthogonal linear projection by minimizing $\text{Tr}(\mathbf{U}\mathbf{X}^\top\mathbf{L}\_w \mathbf{X}\mathbf{U}^\top))$. Kindly note this is non-contrastive learning that only uses $\mathbf{L}\_w$.
>
>   * We define COLES-LPP as minimizing $\text{Tr}(\mathbf{U}\mathbf{X}^\top\mathbf{L}_t \mathbf{X}\mathbf{U}^\top))-\text{Tr}(\mathbf{U}\mathbf{X}^\top\mathbf{L}_w \mathbf{X}\mathbf{U}^\top))$.
>
>   * COLES* minimizes $\text{Tr}(f_\Theta(\mathbf{X})^\top\mathbf{L}\_t f_\Theta(\mathbf{X}))-\text{Tr}(f_\Theta(\mathbf{X})^\top\mathbf{L}\_w f_\Theta(\mathbf{X}))$, where * means two MLP layers are added (as in the DSC model above) are used (added to parameters $\Theta$) between the S$^2$GC backbone and the loss.
>
>   * GLEN-LPP is defined as $\text{rank}(\mathbf{U}\mathbf{X}^\top\mathbf{L}_t \mathbf{X}\mathbf{U}^\top))-\text{rank}(\mathbf{U}\mathbf{X}^\top\mathbf{L}_w \mathbf{X}\mathbf{U}^\top))$.
>
>   * GLEN $\text{rank}(f_\Theta(\mathbf{X})^\top\mathbf{L}\_t f_\Theta(\mathbf{X}))-\text{rank}(f_\Theta(\mathbf{X})^\top\mathbf{L}\_w f_\Theta(\mathbf{X}))$ also uses two MLP layers as in the DSC model.
>
>   * Below are the results:
>     |                           | Cora (5)  | Cora (20) | Citeseer (5) | Citeseer (20) |  Pubmed (5)  | Pubmed (20) | Cora-full (5) | Cora-full (20) |
>     |---------------------------|-----------|-----------|--------------|---------------|-----------|-----------|--------------|---------------|
>     |S$^2$GC | 71.4±4.4 | 81.3±1.2 | 60.3±4.0 | 69.5±1.2 | 67.6±4.2 | 73.3±2.0 | 41.8±1.7 | 60.0±0.5 |
>     | LPP                 | 34.5±1.6  | 54.4±1.5  | 30.5±1.4     | 42.3±1.5     | 39.4±5.3  | 43.9±4.7  | 50.8±1.4     | 61.8±0.5      |
>     | DSC                 | 32.5±3.9  | 53.4±4.6  | 37.2±4.0     | 48.24±3.0      | 40.0±5.6  | 39.2±5.6  | 50.04±0.0     | 60.0±1.0      |
>     | COLES-LPP | 75.0±3.4 | 81.0±1.3 | 67.9±2.3 | 71.7±0.9 | 62.6±5.0 | 73.2±2.6 | 47.6±1.2 | 59.2±0.5      |
>     | COLES* | 73.7±3.0 | 80.4±1.0 | 67.4±2.0 | 71.9±0.9 | 60.3±6.0 | 65.9±1.7 | 23.0±1.4 | 38.3±1.1|
>     | GLEN-LPP                 | 75.3±3.6  | 82.6±1.2  | 65.9±2.7     | 71.5±1.0      | 68.9±3.9  | 78.4±2.1  | 51.4±1.4     | 62.0±0.6      |
>     | GLEN                 | **78.2±2.4** | **83.0±1.0** | **69.1±2.1** | **72.3±0.9** | **70.6±3.9** | **80.1±1.9** | **53.0±1.5** | **62.6±0.5**|
>
>   * Although LPP is a dimensionality reduction method, it significantly weakens the performance of S$^2$GC. In DSC, performance is further degraded by due to MLP.
>
>   * The contrastive term help COLES get better results compared with the baseline S$^2$GC. However, COLES with MLP (COLES*) looses the performance compared with COLES-LPP, e.g., in Cora-full.
>   * In contrast, GLEN-LPP and especially GLEN (which includes MLP) work better than the corresponding competitors, e.g., COLES and COLES*.

---

> ### Author Response · Authors · 2022-08-02
> **Response to Rev. 1 (bffF) (part II of IV)**
>
> # Response to Rev. 1 (bffF) (part II of IV)
>
> ## 2. Compared with the trace model in COLES, the generalization of the proposed Rank difference framework is not illustrated clearly. The author should give proof how the Trace model can be generalized into the Rank model as a special case.
>
>
> * Thank you. In Sec. 5.1 (Eq. 6), we discuss that the nuclear norm $||\cdot||\_*$ used by COLES can be regarded as the $\mathcal{l}\_1$ norm over singular values.
>
> * Below we demonstrate the relationship among the LogDet, Trace and Rank operators, respectively, under the Schatten norm framework. Essential is the following family of objective functions,
> $$
> f_{\alpha,\gamma}(\mathbf{S})=\frac{1}{c}\sum_{i=1}^{d}\log \left(\alpha \sigma_{i}(\mathbf{S})+\gamma\right)=\log \text{det} \left(\alpha\mathbf{S}+\gamma I\right), \quad \alpha, \gamma \geq 0,
> $$
> where $\sigma_{i}(\mathbf{S}), i=1, \ldots, d$, are the eigenvalues of either $\mathbf{S}\_t \in \mathbb{S}\_+^{d}$ or $\mathbf{S}\_w \in \mathbb{S}\_+^{d}$, which are the total scatter matrix and the within scatter matrix from our experiments, respectively. Moreover, $\mathbb{S}\_+^{d}$ is a set of symmetric (semi)definite positive matrices of size $d\times d$ and we define a normalization constant $c$, that is, $c=1$ or $c=\log(\alpha+\gamma)$ as detailed below.
>
> * The relationship between our LogDet function and the Schatten norm is:
> $$
> \lim_{p \rightarrow 0} \frac{S^p_{\gamma, p}(\mathbf{S})-d}{p}= f_{1,\gamma}(\mathbf{S}), \quad \text{where} \quad S_{\gamma, p}(X)=\left(\sum_{i=1}^{d}\left(\sigma_{i}(\mathbf{S})+\gamma\right)^{p})\right)^{1/p}, %, \quad 0<p \leq 1.
> $$
> where $c=1$.
>
> * From the asymptotic analysis, **we can conclude that the LogDet is arbitrarily accurate rational approximation** of $\mathcal{l}_0$ (the so-called pseudo-norm counting non-zero elements) over the eigenvalues of $\mathbf{S}$.
>
> * The **case $p=1$ yields the nuclear norm (Trace) which makes the `smoothed' rank difference of GLEN become equivalent of COLES. The opposing limit case, denoted as $p=0$ recovers LogDet formula**.
>
> * **One can also recover the exact Rank** from the LogDet formulation by:
> $$
> \lim_{\alpha \rightarrow \infty} f_{\alpha,1}(\mathbf{S})=\text{Rank}(\mathbf{S}) \quad \text{if} \quad c=\log(1+\alpha).
> $$
> This is apparent because:
> $$
> \lim_{\alpha \rightarrow \infty} \frac{\log(1+\alpha\sigma_i)}{\log(1+\alpha)} =1 \quad \text{if} \quad \sigma_i>0 \quad \text{and} \quad \lim_{\alpha \rightarrow \infty} \frac{\log(1+\alpha\sigma_i)}{\log(1+\alpha)} =0 \quad \text{if} \quad \sigma_i=0.
> $$
>
>
>
> ## 3. Does the LogDet model still maintain the generalization property?
>
> Yes, if we understood correctly the reviewer's question. Kindly **see Resp. 2, where we show how to recover the Trace based model, and the Rank based model from the LogDet formulation**. Kindly also note we propose in fact the Rank formulation (see Condition 1) as the most general case because in **Theorem 1 (main paper), we offer the expression for the minimum separation between any two embeddings from two different classes (best minimum margin)**. As LogDet model can approach Rank model, in theory, this is the limit on the best separation it can achieve.

---

> ### Author Response · Authors · 2022-08-02
> **Response to Rev. 1 (bffF) (part I of IV)**
>
> # Response to Rev. 1 (bffF) (part I of IV)
>
> *Firstly, **we thank the reviewer** for the constructive review and valuable questions.*
>
> ## 1. What is the relationship between GLEN and contrastive learning?
>
> * COLES [45] extends Laplacian Eigenmaps,  $\min_{{\bf X}, s.t. \Omega({\bf X})} {W}^+\_{ij}\||{\bf x}\_i-{\bf x}\_j\||\_2\^2$ where ${\bf X}=[{\bf x}\_1,\cdots,{\bf x}\_n]$ and $\Omega({\bf X})$ are constraints (i.e., orthogonality) by expanding the SampledNCE formulation $\mathbb{E}\_{i \sim p\_{d}}\left[ \mathbb{E}\_{j \sim p\_{d}(j \mid i)} [s\_{\Theta}(x\_i, x\_j)] + \eta\\, \mathbb{E}\_{j^{\prime} \sim p\_{n}\left(j^{\prime} \mid i\right)} [\tilde{s}\_{\Theta}(x\_i, x\_{j'})]\right]$. Symbols $p\_n$ and $p\_d$ are negative/positive sampling distributions, $s\_{\Theta}(v, u) = \log\exp({\bf u}^{\top} {\bf v})={\bf u}^{\top} {\bf v}$  and $\tilde{s}_{\Theta}(v, u')=\log\exp(-{\bf u}'^\top{\bf v})=-{\bf u}'^\top{\bf v}$ are similarity measures, whereas $\eta\geq 0$ controls the impact of negative sampling.
>
> * **GLEN generalizes SampledNCE**, a framework for contrastive learning with positive and negative sampling, which relies on two terms: $\mathbb{E}\_{v \sim p\_{d}(v)}\left[\mathbb{E}\_{u \sim p\_{d}(u \mid v)} s({\bf u}, {\bf v})\right]$ and $\eta\\,\mathbb{E}\_{v \sim p\_{d}(v)}\left[\mathbb{E}\_{u'\sim p\_{n}\left(u'\mid v\right)} \tilde{s}({\bf u}',{\bf v})\right]$.
>
>   The above two terms are evaluated over two different distributions  $u \sim p_{d}(u \mid v)$ (nodes $u$ from the adjacency matrix) and $u'\sim p_{n}\left(u'\mid v\right)$ (nodes $u'$ from random negative adjacency matrix).
>
>   Take positive sampling term (negative sampling term can be expanded in the similar way). Let $p\_{d}(v) = \frac{1}{\sqrt{D\_{vv}}}$ and $p_{d}(u \mid v) = \frac{\hat{W}\_{uv}}{\sqrt{D\_{uu}}}$ where $\hat{{\bf W}}$ is an unnormalized adjacency matrix and ${\bf D}$ is its degree matrix. Let ${\bf W}$ be the degree normalized matrix. Notice $u$ and $v$ are indexes of embeddings ${\bf u}$ and ${\bf v}$. Let $s$ be as in COLES. Then:
>   $$
>   \mathbb{E}\_{v \sim p\_{d}(v)}[\mathbb{E}\_{u \sim p\_{d}(u \mid v)}\\,s({\bf u},{\bf v})] =\sum\_{u, v} {W}_{uv}\\,s({\bf u},{\bf v}) =\sum\_{i=1}^{d} \sum\_{u, v} {W}\_{uv}u_i v_i= \phi({\bf X}^\top{\bf W}{\bf X}),
>   $$
>   where $\phi(\cdot)$ is a pooling function, i.e., $\phi({\bf M})=\text{Tr}({\bf M})$ yields COLES:
>   $$
>   \sum\_{i=1}^{d}\sum\_{j=1}^{d}\delta(i-j)\phi({\bf x}\_i^\top{\bf W}{\bf x}\_j)=\sum\_{i=1}^{d}\phi({\bf x}\_i^\top{\bf W}{\bf x}\_i) \quad \text{if}\quad {\bf x}\_i\perp{\bf x}\_j\\;\text{for}\\; i\neq j,
>   $$
>   where ${\bf x}\_i\perp{\bf x}\_j\$ imposes orthogonality constraints of Laplacian eigenmaps and $\delta(z)=1$ if $z=0$ and $\delta(z)=0$ if $z\neq0$. Finally ,think that rows of ${\bf X}$ contain all ${\bf u}$ (and ${\bf v}$).
>
>   **We let the pooling operator $\phi(\cdot)$ operate on the entire spectrum under general aggregation scheme. A very general operator is $\phi({\bf M})=\text{Rank}({\bf M)}$ from which we can recover the original Trace (nuclear norm) of COLES or LogDet of GLEN, $\gamma$-nuclear, $S\_p$, and Geman norm.**
>
> * COLES uses the following expression based on SampledNCE:  $\min_{{\bf X}} \sum_{ij} {W}^+\_{ij}\||{\bf x}\_i-{\bf x}\_j\||\_2\^2- (\frac{\eta}{\kappa}\sum\_{l=1}\^\kappa{W}^{l,-}\_{ij})\||{\bf x}\_i-{\bf x}\_j\||\_2\^2 = \max_{{\bf X}}\text{Tr}({\bf X}^\top{\bf L}\_t{\bf X})-\text{Tr}({\bf X}^\top {\bf L}\_w{\bf X})$, where the ${\bf W}^+$ is a normalized adjacent matrix and ${\bf W}^{l,-}$ are $\kappa$ normalized randomized $k$-regular graphs (adjacent matrices), while ${\bf L}\_w$ and ${\bf L}\_t$ are the corresponding Laplacian matrices.
>
> * **Negative random sampling is represented by**  ${\bf W}^{-}$, e.g., randomized $k$-regular graph or several such graphs.
>
> * If we sample $\kappa\rightarrow\infty$ randomized $k$-regular graphs (adjacent matrices of size $n\times n$) (each row receives 1 with probability $k/n$), **the expectation of randomized graph (adjacent matrix) is** $\mathbb{E}[{\bf W}^-] =\lim_{\kappa\rightarrow\infty}\frac{1}{\kappa}\sum\_{l=1}\^\kappa{{\bf W}}^{l,-}= \frac{k}{n}{\bf 1}{\bf 1}^\top$, which by itself is a fully-connected graph with the graph Laplacian ${\bf L}\_t={\bf I}-\frac{k}{n}{\bf 1}{\bf 1}^\top$.
>
> * **We simply set $k=1$ to use $1$-regular graphs for negative sampling** so ${\bf L}\_t={\bf I}-\frac{1}{n}{\bf 1}{\bf 1}^\top$. Thus, our contrastive term is equivalent of the total scatter matrix ${\bf S}_t$ known from the Linear Discriminant Analysis, i.e., ${\bf S}_t={\bf X}^\top({\bf I}-\frac{1}{n}{\bf 1}{\bf 1}^\top){\bf X}={\bf X}^\top {\bf L}\_t{\bf X}$. The positive sampling is encoded by the graph adjacency matrix $ {\bf L}\_w$.
>
> * Thus, our GLEN is given as:
>   $$
>   \max_{{\bf X}} \text{Rank}({\bf X}^\top {\bf L}\_t{\bf X})-\text{Rank}({\bf X}^\top {\bf L}\_w{\bf X})
>   $$

---

> > ### Public Comment · ~Noboru_Isobe1 · 2022-12-06
> > **Is the definition of $L_w$ inconsistent?**
> >
> > Thank you for sharing about the exciting relationship between GLEN and contrastive learning. I want to ask you a few questions to understand this relationship better. Your reply above stated that $\displaystyle L_w(=\mathbf{I}-\mathbf{W}^+)$ is a Laplacian matrix. On the other hand, from equation (1) in section 3.1, $\displaystyle L_w=\mathbf{I}-\sum_{c=1}^C \frac{1}{n_c} \mathbf{e}^c \mathbf{e}^{c \top}$. Hence, we obtain $\displaystyle\mathbf{W}^+=\sum_{c=1}^C \frac{1}{n_c} \mathbf{e}^c \mathbf{e}^{c \top}$. This result means that the adjacency matrix can be decomposed by a matrix of the form $\mathbf{e}^c \mathbf{e}^{c \top}$. However, this is unreasonable since the diagonal components of the adjacency matrix $\mathbf{W}^+$ are zero, whereas the components of $\mathbf{e}^c \mathbf{e}^{c \top}$ have non-zero elements. I need to correct the definition of equation (1). How should I fix it?

---

> > > ### Public Comment · Authors · 2022-12-28
> > > **Remarks on adjacency.**
> > >
> > > Thank you for your question.
> > >
> > > In our paper, we define a generalized Laplacian matrix $\displaystyle \mathbf{L}_w=(\mathbf{I}-\mathbf{W}^+)$ where $\mathbf{W}^+$ could be a given adjacency matrix (from graph datasets) or a constructed adjacent matrix based on a distance between labels or features.
> > >
> > > In our Eq.1 we define $\mathbf{W}^+=\sum_{c=1}^C \frac{1}{n_c} \mathbf{e}^c \mathbf{e}^{c \top}$ to represent the within-class matrix as a Laplacian matrix (can be deformed Laplacian etc.) for for the sake of understanding the further definition.
> > >
> > > We are not clear about the motivation of this question or how/why one would like to modify the definition of Eq.1 as Eq. 1 is never used in the actual problem solver. Eq. 1 is used for the analysis and understanding of the model.
> > >
> > > So we list some possible ways of how the adjacency matrix could be defined for Eq. 1.
> > >
> > > 1. One may use an adjacency matrix $\mathbf{W}$ where the diagonal elements are zero.
> > > 2. One does not have an adjacent matrix $\mathbf{W}$ given directly but one constructs it from:
> > >     - features so one has to choose a kernel function (RKHS RBF, etc.) to estimate a distance between all pairs $\mathbf{x}_i$ and $\mathbf{x}_j$ where $\mathbf{x}_i$ could be for example the feature vector or the attribute vector of the $i$-th node. Then one can construct an adjacency matrix with zeroes along the diagonal elements.
> > >     - labels if one has labels as then one could construct the adjacency matrix from the definition of our Eq.1 exactly (although one may think of it as a deformed adjacency).
> > >
> > > Regarding the relation to the within-class matrix $\mathbf{X}'\mathbf{L}_w\mathbf{X}$, one can remove diagonal elements and then it is not an exact within-class matrix but some approximation to it: $\mathbf{X}'\mathbf{L}_w\mathbf{X}-\mathbf{X}'(\sum\_{c=1}^C \frac{1}{n_c}diag(\mathbf{e}\^c))\mathbf{X}$ where $diag(\cdot)$ converts vector $\mathbf{e}^c$ into a diagonal matrix.
> > >
> > > Kindly note our Eq.1 only illustrates the relationship between within-class, between-class and the total scatter matrices. We do not use them for the graph embedding as the adjacency matrix is given by the datasets.

---

### Author Response · Authors · 2022-08-09
**We are here to answer any questions.**

We thank the reviewers and the AC for their work.

As the reviewer-author discussion period is finishing in the next few hours, we just wanted to say that we are here to help should you have any additional questions.

Best regards,
Authors.

---

### Meta-Review · Area_Chair_GJhu · 2022-08-25

**Recommendation:** Accept
**Confidence:** Certain

**Metareview:**

The Authors provided a nice rebuttal, and address major issues in the last round. Therefore, I  recommend to accept this paper.

**Award:**

No

---

### Decision · Program_Chairs · 2022-09-14

Accept